# SARS-CoV-2 viral entry and replication is impaired in Cystic Fibrosis airways due to ACE2 downregulation

Valentino Bezzerri [1,2], Valentina Gentili [3], Martina Api[4], Alessia Finotti[5,6], Chiara Papi [5], Anna Tamanini[7], Christian Boni [1], Elena Baldisseri [1], Debora Olioso[2], Martina Duca[4], Erika Tedesco[1], Sara Leo[8], Monica Borgatti [5,6], Sonia Volpi[1], Paolo Pinton [6,8], Giulio Cabrini[5,6], Roberto Gambari[5,6], Francesco Blasi [9,10], Giuseppe Lippi[2], Alessandro Rimessi[6,8,11], Roberta Rizzo [3,11] & Marco Cipolli [1,6,11] ✉

As an inherited disorder characterized by severe pulmonary disease, cystic fibrosis could be considered a comorbidity for coronavirus disease 2019. Instead, current clinical evidence seems to be heading in the opposite direction. To clarify whether host factors expressed by the Cystic Fibrosis epithelia may influence coronavirus disease 2019 progression, here we describe the expression of SARS-CoV-2 receptors in primary airway epithelial cells. We show that angiotensin converting enzyme 2 (ACE2) expression and localization are regulated by Cystic Fibrosis Transmembrane Conductance Regulator (CFTR) channel. Consistently, our results indicate that dysfunctional CFTR channels alter susceptibility to SARS-CoV-2 infection, resulting in reduced viral entry and replication in Cystic Fibrosis cells. Depending on the pattern of ACE2 expression, the SARS-CoV-2 spike (S) protein induced high levels of Interleukin 6 in healthy donor-derived primary airway epithelial cells, but a very weak response in primary Cystic Fibrosis cells. Collectively, these data support that Cystic Fibrosis condition may be at least partially protecting from SARS-CoV-2 infection.

The genome of severe acute respiratory syndrome coronavirus 2 (SARS-CoV-2) encodes 28 proteins, including four structural proteins: spike (S), membrane, envelope and nucleocapsid. The S glycoprotein of SARS-CoV-2 is responsible for viral entry through the binding to the Angiotensin Converting Enzyme 2 (ACE2) receptor[1,2], which is ubiquitously distributed in different tissues, remarkably in pulmonary epithelial cells and intestinal enterocytes[3]. Once the S protein has bound the ACE2 receptor, it is processed by several proteases, including transmembrane protease serine 2 (TMPRSS2) and furin, which promote priming of S protein and the fusion of viral and cellular membranes[1].

[1]Cystic Fibrosis Center of Verona, Azienda Ospedaliera Universitaria Integrata, Verona, Italy. [2]Section of Clinical Biochemistry, University of Verona, Verona, Italy. [3]Department of Chemical and Pharmaceutical Sciences, University of Ferrara, Ferrara, Italy. [4]Cystic Fibrosis Center of Ancona, Azienda Ospedaliero Universitaria Ospedali Riuniti, Ancona, Italy. [5]Department of Life Sciences and Biotechnology, University of Ferrara, Ferrara, Italy. [6]Center on Innovative Therapies for Cystic Fibrosis, University of Ferrara, Ferrara, Italy. [7]Section of Molecular Pathology, Azienda Ospedaliera Universitaria Integrata, Verona, Italy. [8]Department of Medical Sciences, University of Ferrara, Ferrara, Italy. [9]Internal Medicine Department, Respiratory Unit and Cystic Fibrosis Center, Fondazione IRCCS Cà Granda Ospedale Maggiore Policlinico Milano, Milan, Italy. [10]Department of Pathophysiology and Transplantation, Università degli Studi di Milano, Milan, Italy. [11]These authors contributed equally: Alessandro Rimessi, Roberta Rizzo, Marco Cipolli. ✉e-mail: marco.cipolli@aovr.veneto.it

In some cases, infection leads to bilateral pneumonia with diffuse alveolar damage, which in turn may promote acute respiratory distress syndrome (ARDS), especially in subjects with comorbidities. ARDS has been closely related to the cytokine storm, also referred as cytokine release syndrome (CRS)[4,5]. CRS has already been described as a critical factor for severe outcomes also in SARS-CoV(−1) and MERS-CoV infections[4]. Interleukin (IL)−6 plays a key role in CRS and its plasma values correlates with COVID-19 severity[6,7].

Cystic Fibrosis (CF) is caused by mutations in the Cystic Fibrosis Transmembrane conductance Regulator (*CFTR*) gene, which encodes a chloride and bicarbonate channel widely expressed in human epithelia. Loss of *CFTR* expression or function in airway epithelia is associated with reduced airway surface liquid (ASL) volume and dehydration. This process has been suggested as the initiating event of CF airway disease pathogenesis, which is characterized by severe impairment of lung function[8]. CF could therefore be considered to be an unfavorable comorbidity in patients with COVID-19, particularly considering that other respiratory viral infections, including respiratory syncytial virus (RSV) and influenza A (H1N1), lead to a rapid deterioration of lung function and increased mortality in CF patients[9,10].

However, several studies conducted on Belgian[11], French[12], Spanish[13], German[14], and Italian[15] cohorts of CF patients have reported that they generally exhibit mild symptoms upon SARS-CoV-2 infection. Even though age distribution has been proposed as a major confounding factor for incidence calculation in these studies, the European Society of Cystic Fibrosis (ECFS) recently concluded that the case fatality rate associated with COVID-19 in CF patients was lower than that calculated in the general population[16]. On the other hand, it should not be dismissed that post lung transplant patients exhibited more severe manifestations in response to COVID-19[17,18].

Since 51.2% of patients registered in the ECFS patient registry are adults[16], the possibility that the favorable COVID-19 disease outcome could be dependent only on pediatric age is questionable.

Although behavior factors, including improved ability and motivation to self-isolate, may contribute to the reduced number of critical COVID-19 cases in patients with CF, our hypothesis was that some specific host factors associated with CF may influence susceptibility to SARS-CoV-2 infection. Interestingly, it has been recently reported that TMEM16F, a $Ca^{2+}$-activated chloride channel, plays a key role in SARS-CoV-2 viral entry and syncytia formation in lung epithelial cells[19], being actively involved in the pathogenesis of COVID-19. Furthermore, it has already been established that CFTR can regulate other apical proteins, including the ion channel solute carrier family 26 member 9 (SLC26A9)[20], the epithelial sodium channel (ENaC)[21], the potassium channel KIR 1.1[22], Phosphatase and Tensin Homolog protein (PTEN)[23], and receptors such as the A2B adenosine receptor[24]. Regulatory effect of CFTR on other ion channels may be explained by its impact on total cell potential difference, thus as a unifying mechanism for many other ion channels. Nevertheless, these regulatory functions of the CFTR might be due to direct binding of channel to other proteins or indirect binding through PDZ-interacting domains through the adapter proteins ezrin, $Na^+/H^+$ exchanger regulatory factor (NHERF)1/2. Interestingly, it has been recently reported that NHERF-1 may directly interact with ACE2 through the PDZ-binding motif in human lung and intestinal cells, thus facilitating SARS-CoV-2 viral entry[25].

Given all of these possibilities, we investigated the role of CFTR in regulating SARS-CoV-2 receptor, namely ACE2. We found that both ACE2 expression and localization correlated with CFTR expression and localization. Consistently, our results indicate that SARS-CoV-2 viral entry and replication are significantly reduced in CF cells. Indeed, we found that SARS-CoV-2 S protein is unable to induce high levels of IL-6 release in CF primary airway epithelia, whereas it promotes a considerable release of IL-6 in primary cultures derived from healthy donors.

## Results

### ACE2 expression is downregulated in CF primary airway epithelia

*ACE2* mRNA expression was significantly reduced in both primary human bronchial epithelial cells (hBEC) and nasal epithelial cells (hNEC) isolated from CF patients (Supplementary Table 1), compared to healthy donor-derived tissues (Fig. 1a, b). Consistent with the mRNA data, ACE2 protein levels were remarkably lower in primary well-differentiated CF-hNEC and CF-hBEC (Fig. 1c–f), resulting in 25% and 38% of the healthy control tissues, respectively.

In order to verify whether the function of CFTR channel is involved in this process, we incubated primary hBEC with the well-established thiazolidinone inhibitor CFTR(inh)−172[26]. Our results indicated that ACE2 expression was not affected by the inhibition of CFTR chloride efflux in hBEC (Fig. 1d, f).

### CFTR expression positively correlates with ACE2 protein levels in bronchial epithelial cells

To determine whether *CFTR* expression influences *ACE2* expression, we utilized different bronchial epithelial cell models in which gene editing approaches were applied to modulate *CFTR* expression. In particular, we employed the human lung adenocarcinoma Calu-3 cell line in which *CFTR* was stably silenced by a short hairpin (sh)RNA (SH3 cells), as previously described[27], the well-established CF bronchial epithelial cell line CFBE41o- (parental), in which *CFTR* expression is almost undetectable (null), and cell lines derived from CFBE41o- cells over-expressing wild-type *CFTR* (CFBE41o- WT) or F508del-*CFTR* (CFBE41o- F508del). Finally, we used human bronchial epithelial 16HBE14o- cell line, typically considered a normal control, although this cell model has been shown to be heterozygous for wild-type *CFTR*, due to the insertion of the SV40 sequence within one of the two *CFTR* alleles[28]. This cell line was subsequently edited using the CRISPR/Cas9 approach to obtain two additional cell lines carrying biallelic W1282X-*CFTR* or G542X-*CFTR* nonsense mutations.

The results unambiguously indicated that *CFTR* expression is associated with *ACE2* expression. In particular, *CFTR*-deficient SH3 cells displayed a reduction of almost 18% and 19% of ACE2 levels compared to parental Calu-3 cells and mock-transfected cells (i.e., Alter[27] cells), respectively (Fig. 2a–c). Then we assessed *ACE2* expression in W1282X- and G542X-*CFTR* expressing 16HBE14o- cell lines, which show undetectable levels of CFTR protein (Supplementary Fig. 1a, b). Consistently with the results from SH3 and Alter cells, we found that polarized 16HBE14o- cells, harboring W1282X- and G542X-*CFTR*, showed 10% and 19% of ACE2 protein levels compared with parental 16HBE14o- cells (Fig. 2d, e). Similar results were obtained employing unpolarized cell lines (Supplementary Fig. 1a, c). According to what has been observed in primary hBEC, incubation of polarized 16HBE14o- cells with CFTR(inh)−172 did not influence the expression level of ACE2 protein (Fig. 2d, e). Moreover, the over-expression of both mutated F508del-*CFTR* and wild-type *CFTR* in polarized *CFTR*-null CFBE41o- parental cells[29] resulted in a considerable increase in ACE2 levels (Fig. 2f, g). Again, similar results were obtained employing unpolarized cell lines (Supplementary Fig. 1d–f).

### CFTR channel and ACE2 receptor co-localize in bronchial epithelial cells

We investigated whether the subcellular distribution of ACE2 receptor is associated with CFTR protein expression, localization and function, considering that this channel exists in a multiprotein complex that regulates CFTR channel activity but may also regulate the localization and function of other proteins on the plasma membrane[21–24]. For this purpose, we tested different bronchial epithelial cell lines by immunofluorescence using single-cell analysis. In CFBE41o- (null) cells, both CFTR and the ACE2 receptor were not localized on the plasma membrane (Fig. 3a). The typical plasma membrane localization of the ACE2

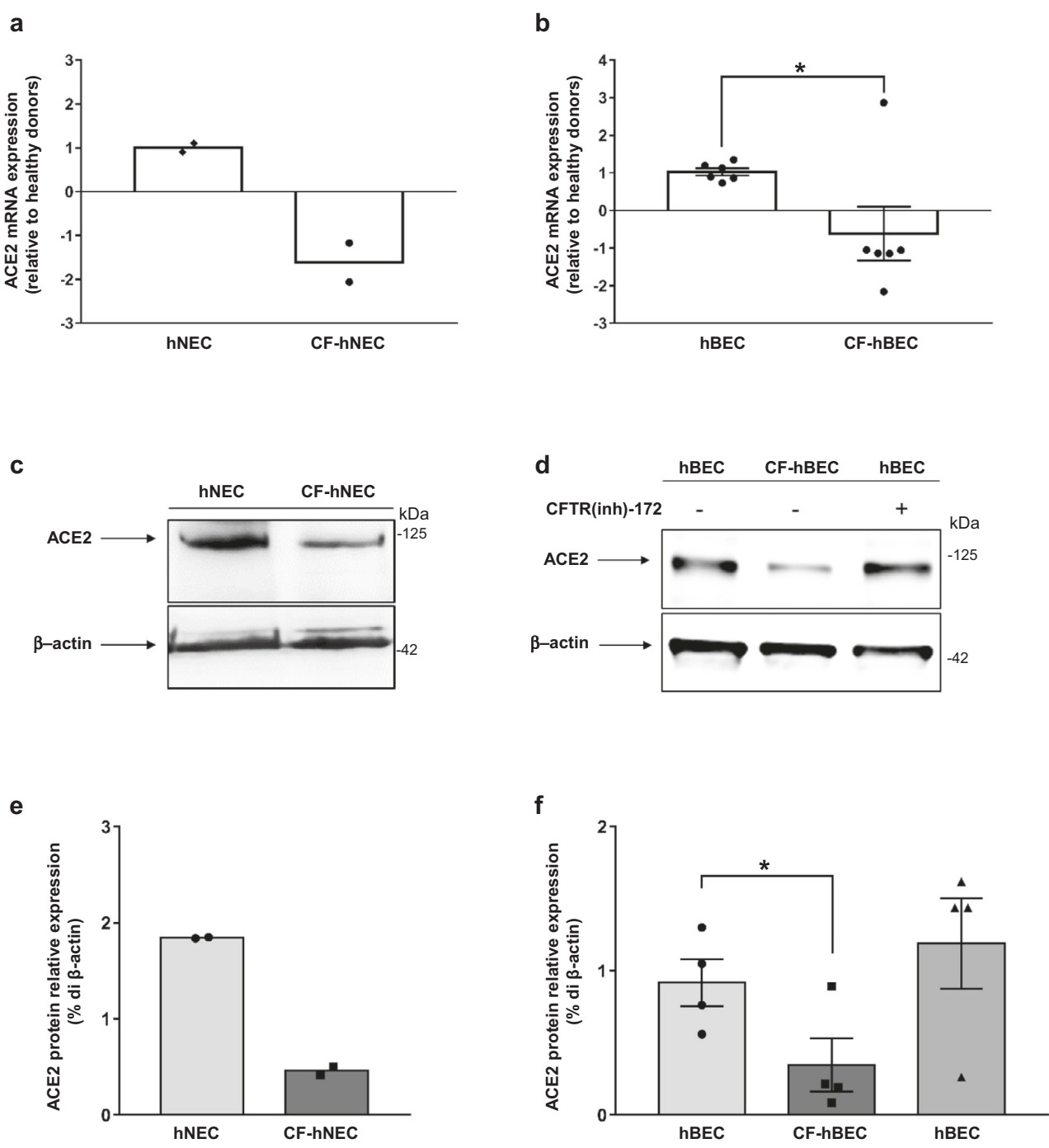

**Fig. 1 | ACE2 expression is reduced in CF airway epithelia. a, b** Quantification of ACE2 mRNA expression by qPCR in lysates collected from well-differentiated hNECs and hBECs, grown at the air-liquid interface. Epithelia were derived from a pool of fourteen healthy donors (hNEC) and two CF patients (CF-hNEC) (**a**), or from six healthy donors (hBEC) and six CF patients (CF-hBEC) (**b**). The genotypes of the patients enrolled in this study are reported in Supplementary Table 1. Data are shown as the mean ± SEM of independent experiments ($n = 4$ and $n = 6$ for hNEC and hBEC, respectively). **c** Representative western blot analysis of protein extracts from differentiated hNECs grown at the air–liquid interface obtained from a pool of fourteen healthy control subjects versus CF-hNECs obtained from two F508del-CFTR homozygous CF patients. **d** Western blot analysis of ACE2 in differentiated hBEC grown at the air–liquid interface obtained from four CF patients compared with four healthy donors. Normal hBEC were additionally incubated in the presence (+) or absence (−) of 5 µM CFTR inhibitor CFTR(inh)-172 for 24 h. **e** Densitometry analysis (% of β-actin expression) of two independent experiments ($n = 2$), conducted as reported in (**c**). **f** Densitometry analysis (% of β-actin expression) of four independent experiments ($n = 4$) performed in protein extracts from hBEC, as reported in panel (**d**). Data represented in panels **e**, **f** are shown as the mean ± SEM. Normal distribution was confirmed using the Shapiro–Wilk test before running two-tailed Student's t-test (**a, b, f**), which has been reported in the scatter plot with bars (*$p < 0.05$). Source data are provided as a Source Data File.

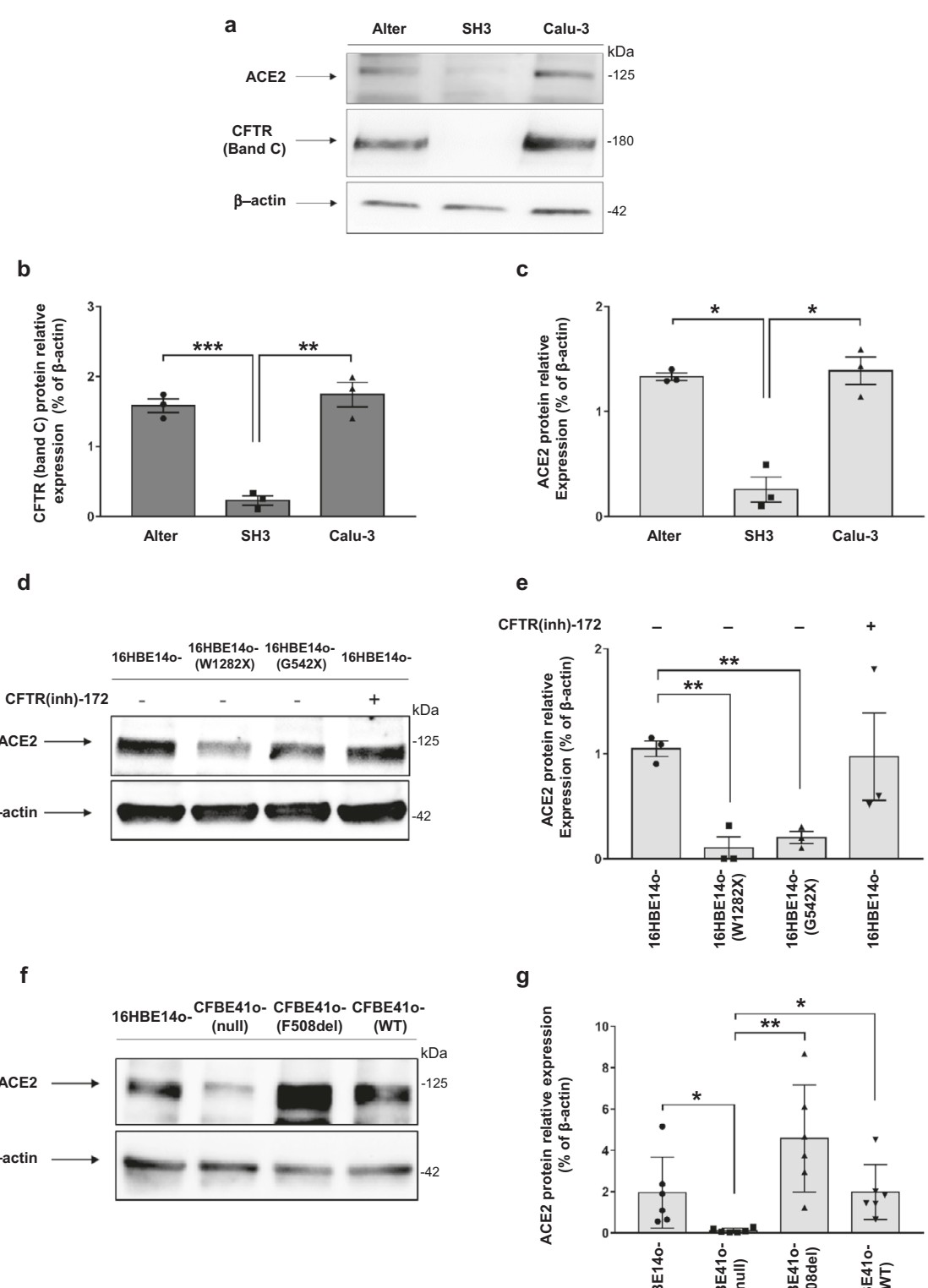

**Fig. 2 | CFTR expression positively correlates with ACE2 protein expression in bronchial epithelial cells. a** Representative western blot analysis of ACE2 and CFTR in Calu-3, SH3 and Alter cells. **b, c** Densitometry analysis of CFTR and ACE2 bands depicted in panel (**a**). Data are shown as the mean ± SEM of three independent experiments (*n* = 3). **d** Normal 16HBE14o- cells and gene-edited 16HBE14o- cells carrying biallelic W1282X-CFTR or G542X-CFTR mutations, grown under polarized conditions, were additionally incubated in the presence (+) or absence (−) of 5µM CFTR inhibitor CFTR(inh)-172 for 24 h. Proteins were extracted and western blot analysis was performed (representative image). **e** Densitometry analysis

conducted in western blots as represented in panel (**d**). Data are shown as the mean ± SEM of three independent experiments (*n* = 3). **f** Western blot analysis on polarized CFBE41o- cells (null) and cells over-expressing the F508del- or wild-type (WT) *CFTR* (representative image). **g** Densitometry analysis conducted in western blots as represented in panel (**f**). Data are shown as the mean ± SEM of three independent experiments (*n* = 3). Normal distribution was confirmed using the Shapiro−Wilk test before running two-tailed Student's t-test for paired data (**b, c, e, g**). (*$p < 0.05$; ** $p < 0.01$; ***$p < 0.001$). Source data are provided as a Source Data File.

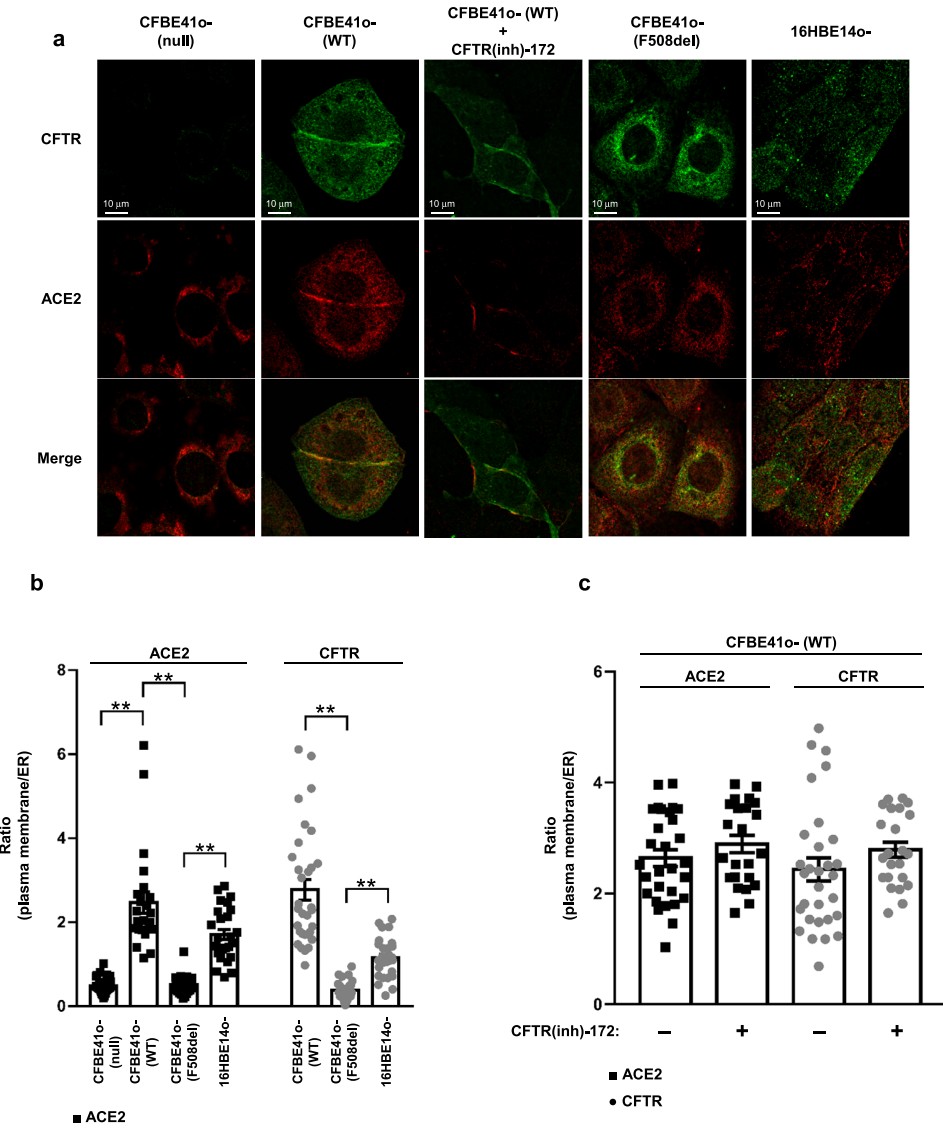

**Fig. 3 | Localization of CFTR on the plasma membrane, but not its function, is essential for the subcellular localization of ACE2 on the cell surface.**
**a** Representative images of immunofluorescence detection of CFTR (green) and ACE2 receptor (red) under basal conditions in CFBE41o- (null), CFBE41o- expressing wild-type *CFTR* (WT), or F508del-*CFTR* (F508del), and 16HBE14o- cells. CFBE41o- (WT) cells were incubated also with 5 µM CFTR(inh)-172 for 24 h. Images were acquired with a Zeiss LSM510 confocal microscope (scale bar: 10 µm).
**b** Quantification of the subcellular distribution of the ACE2 receptor (black) and CFTR (gray) in the different cell models. The scatter plot with bars represents the ratio between plasma membrane fluorescence and endoplasmic reticulum (ER) fluorescence. **c** The scatter plot with bars represents the ratio between plasma membrane fluorescence and endoplasmic reticulum (ER) fluorescence of ACE2 (black) and CFTR (gray). Data are mean ± SEM of six independent experiments ($n = 6$). Normal distribution was tested by the Shapiro−Wilk test before running the two-tailed Student's t test for paired data (**b**, **c**), which has been reported within the scatter plot with bars (**p < 0.01). Source data are provided as a Source Data File.

receptor was instead observed in CFBE41o- (WT) cells over-expressing wild-type *CFTR* (Fig. 3a), in which the subcellular distribution of ACE2 on the plasma membrane was strictly associated with CFTR, as confirmed by Manders M1 and Pearson's coefficients (Supplementary Fig. 2a, b). Indeed, these parameters sustain the correlation between the intracellular localization of CFTR and ACE2. Interestingly, in CFBE41o- cells over-expressing the F508del-*CFTR*, which encodes an unfolded protein that is primarily retained in the ER, ACE2 was almost entirely localized into the ER (Fig. 3a). The redistribution of ACE2 to the ER was confirmed by subcellular quantification, since the ratio between plasma membrane fluorescence and ER fluorescence was significantly reduced in CFBE41o- (F508del) cells compared to CFBE41o- (WT) and 16HBE14o- cells, where both CFTR and ACE2 proteins are primarily localized on the plasma membrane (Fig. 3b). Next, we examined whether CFTR function could be involved in ACE2

localization. To this aim, we incubated 16HBE14o- and CFBE41o- (WT) cells with CFTR(inh)−172 for 24 h. Our data indicated that the prolonged functional block of ion efflux from CFTR does not lead to modification of the subcellular localization of ACE2 (Fig. 3c and Supplementary Fig. 2c, d). Similar results were obtained in bronchial epithelial cells grown under polarized conditions. In these models, the colocalization of ACE2 and CFTR on the apical plasma membrane was evident in CFBE41o- (WT) and 16HBE14o- cells (Fig. 4a, b). Furthermore, the complete loss of *CFTR* expression displayed in *CFTR*-null models, including CFBE41o- (null) and W1282X- or G542X-*CFTR* expressing cells, led to an evident reduction of ACE2 levels, in particular on the apical plasma membrane (Fig. 4a, b).

The reduction of ACE2 protein expression and its mislocalization outside the plasma membrane is even more visible in primary well-differentiated bronchial epithelia obtained from CF patients

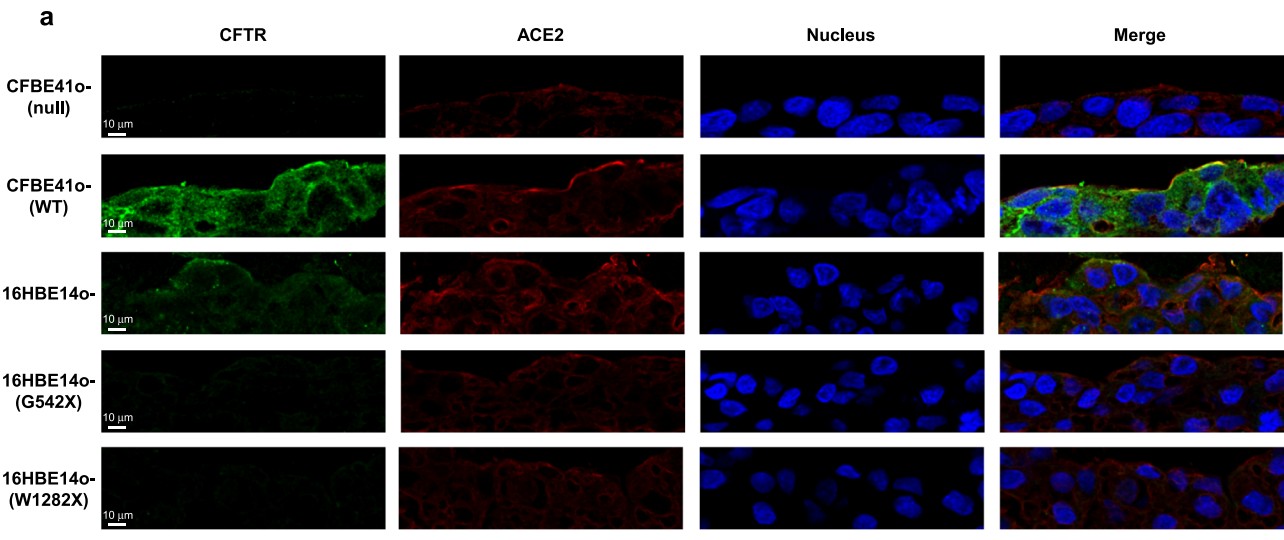

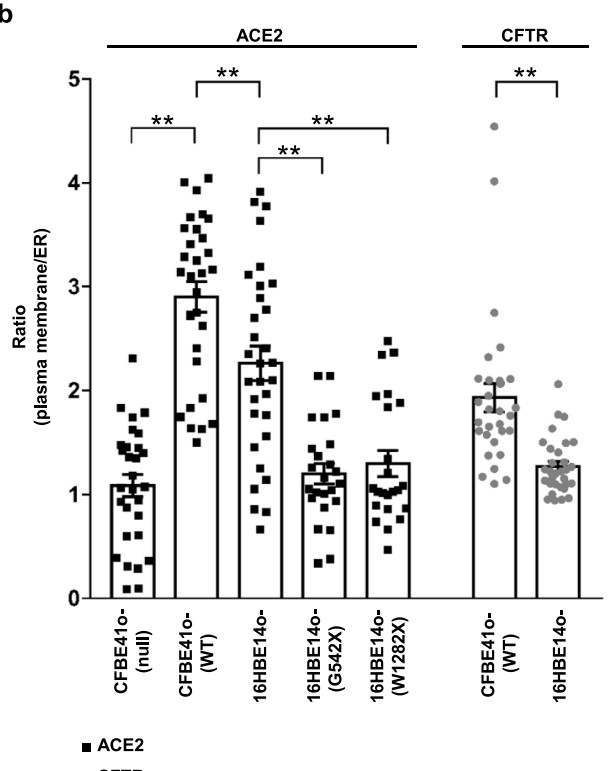

**Fig. 4 | Intracellular distribution of ACE2 and CFTR in polarized bronchial epithelial cells. a** Representative images of immunofluorescence detection of CFTR (green), ACE2 receptor (red) under basal conditions in polarized CFBE41o-(null), CFBE41o- expressing wild-type CFTR (WT), 16HBE14o- cells and in gene-edited 16HBE14o- cells carrying biallelic G542X-CFTR mutations (G542X) or W1282X-CFTR (W1282X), respectively. Images were acquired with confocal laser scanner Olympus FV3000 microscope (scale bar: 10 μm). Hoechst stain was used to fluorescently label the cell nuclei (blue). **b** The scatter plot with bars represents the quantification of the subcellular distribution of the ACE2 receptor (black) and CFTR (gray) in the different cell models, expressed as ratio between plasma membrane fluorescence and endoplasmic reticulum (ER) fluorescence. Data are mean ± SEM of 31 cells examined over 6 independent experiments ($n = 31$). Normal distribution was tested by the Shapiro−Wilk test before running the two-tailed Student's t test for paired data (**b**), which has been reported within the scatter plot with bars (**$p < 0.01$). Source data are provided as a Source Data File.

carrying the F508del or class I mutations in *CFTR* (Fig. 5). In these tissues, correct ACE2 protein expression on plasma membrane was observed only in healthy donor-derived cells (Fig. 4b). In particular, the epithelium obtained from the CF patient with 2184insA /W1282X *CFTR* genotype (CF-MD0208) displayed very low levels of ACE2 (Fig. 5a), even compared with patient CF-MD0673 harboring F508del-*CFTR*.

To exclude the possibility that mislocalization of the ACE2 receptor into the ER is due to alterations in the membrane trafficking pathway associated with the unfolded CFTR protein, we investigated the subcellular localization of CFTR-interactor NHERF1[30]. In the absence of *CFTR* expression, NHERF1 localizes closely to the plasma membrane in CFBE41o- (null) cells (Supplementary Fig. 3a). NHERF1 normally binds to the CFTR channel through its PDZ domains. This

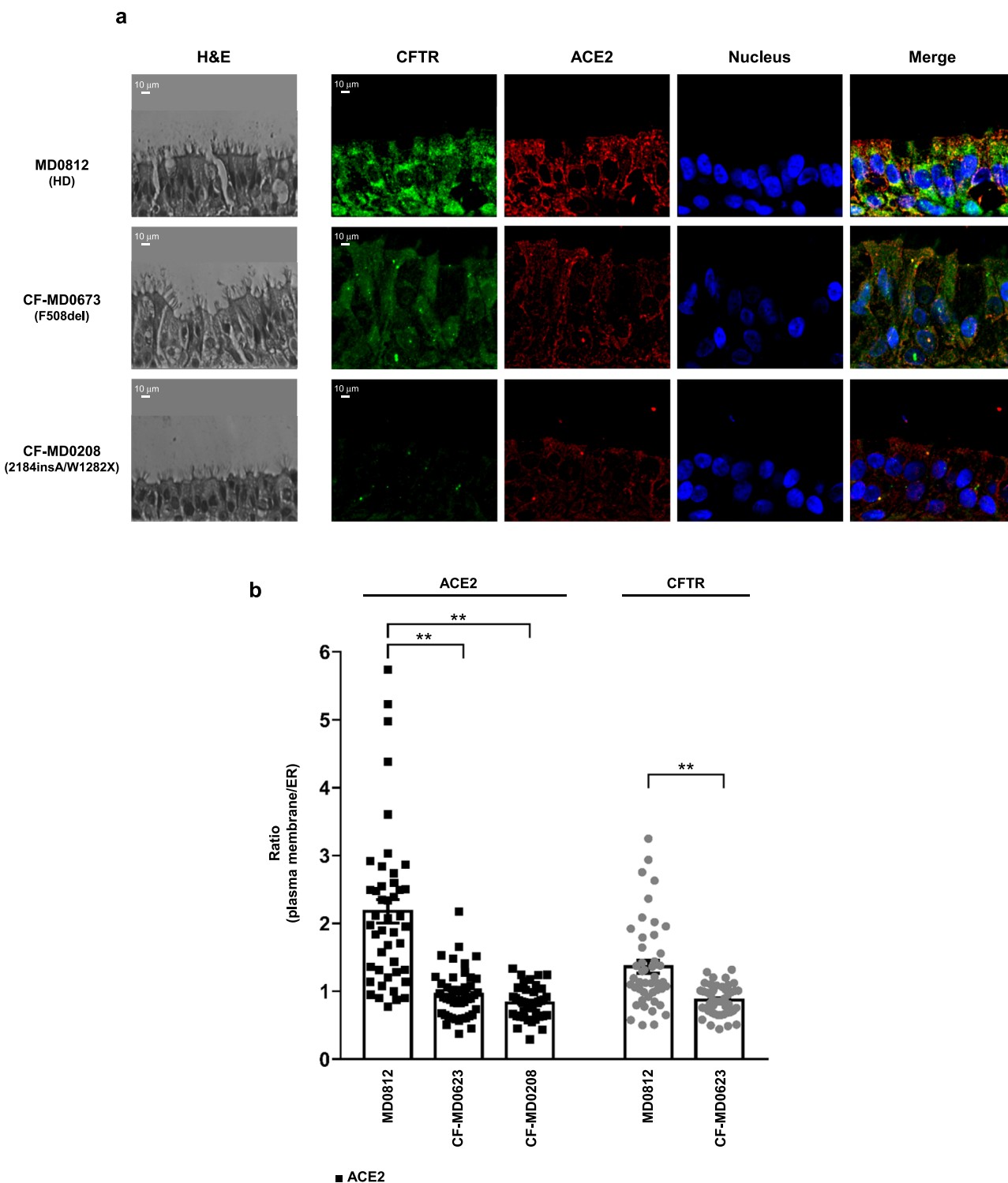

**Fig. 5 | Distribution of ACE2 and CFTR in primary air–liquid interface differentiated hBEC obtained from healthy donors and CF patients.**
**a** Immunofluorescence detection of ACE2 and CFTR in primary well-differentiated hBEC obtained from an healthy donor (HD, MD0812), CF patient homozygous for F508del-*CFTR* mutation (CF-MD0673) and CF patient with genotype 2184insA/W1282X *CFTR* (CF-MD0208). H/E images have been reported to show the cell structure of primary air-liquid interface differentiated hBEC obtained from donor and CF patients. The images were acquired with a confocal laser scanner Olympus FV3000 microscope (scale bar: 10 μm). **b** The histogram represents the ratio between plasma membrane fluorescence and endoplasmic reticulum (ER) fluorescence of ACE2 (black) and CFTR (gray). Data are mean ± SEM of 46 cells examined over 6 independent experiments (*n* = 46). Normal distribution was tested by the Shapiro–Wilk test before running the two-tailed Student's t test (**b**), which has been reported within the scatter plot with bars (**p* < 0.01). Source data are provided as a Source Data File.

interaction was indeed confirmed by the coefficients of colocalization in CFBE41o- (WT) cells re-expressing wild-type *CFTR* (Supplementary Fig. 3b, c). The typical localization close to the plasma membrane of NHERF1 was not perturbed even in CFBE41o- cells expressing the mutated F508del-CFTR channel, which is retained in the ER instead (Supplementary Fig. 3d). These results therefore support the hypothesis that the presence of a mutated CFTR channel, as well as the loss of *CFTR* expression, may induce mislocalization of the ACE2 receptor regardless of the localization of NHERF1, one of the major stabilizers of CFTR interactome.

### Down-modulation of CFTR expression and function reduces SARS-CoV-2 entry and replication

To confirm the role of CFTR in the SARS-CoV-2 life cycle, we employed two complementary strategies, including the use of pharmacological inhibition of CFTR function, namely CFTR(inh)−172, and a miRNA-based approach to specifically downregulate CFTR synthesis, using the Calu-3 cells. We utilized a previously validated methodology for in vitro viral infections[31].

The polarized 16HBE14o- cell line and its mutants (G542X- and W1282X-*CFTR*) were infected with SARS-CoV-2 and analyzed for viral replication. In early time points (i.e., 8 h post infection), we observed elevated viral replication in 16HBE14o- cells compared with mutant cells, which was maintained until 72 h post-infection (Fig. 6a). The viral titration confirmed the RNA results, with total loss of infectivity in G542X- and W1282X-*CFTR* expressing cells in all the tested time points (Fig. 6b).

Regarding the miRNA-based strategy, our group and others have previously demonstrated that *CFTR* expression is under post-transcriptional control of different microRNAs[32–36]. Since *CFTR* expression increased in response to antagomiR molecules against miR-145-5p[32,35,36], the exposure of bronchial epithelial cells to this miRNA might lead to *CFTR* downregulation (see Fig. 6c for localization of miR-145-5p binding sites within the CFTR 3′UTR). In fact, four miR-145-5p binding sites are present in the 3′-UTR of *CFTR*. Treatment of Calu-3 cells with pre-miR-145-5p significantly reduced *CFTR* mRNA levels (Fig. 6d). Most importantly, treatment of Calu-3 cells with pre-miR145-5p remarkably reduced CFTR protein levels (Fig. 6e, f). The conclusion of these experiments fully supports the concept that downregulation of *CFTR* can be achieved by a miRNA-mimicking strategy using pre-miR-145-5p.

We next assessed whether this treatment, performed on SARS-CoV-2-infected Calu-3 cells, is associated with alteration of the SARS-CoV-2 life cycle. The results revealed that pretreatment of Calu-3 cells with pre-miR-145-5p is sufficient to considerably inhibit SARS-CoV-2 viral infectivity (Fig. 6h) and replication (Fig. 6g–i) in a similar fashion to that observed by incubating Calu-3 cells with the CFTR(inh)-172 compound, which does not affect CFTR processing (Fig. 6g−i). Check of CFTR protein expression upon pre-miR-145-5p and CFTR(inh)-172 treatments in samples tested for viral entry is shown in Fig. 6j, k.

### Reduced ACE2 protein level observed in CF cells is associated with decreased SARS-CoV-2 Spike recognition and subsequent IL-6 release

The evident reduction of ACE2 levels observed in CF airway epithelia is expected to result in reduced SARS-CoV-2 S protein recognition. In order to address this point, we performed a proximity ligation assay in CFBE41o- cells. Data confirmed that the binding of SARS-CoV-2 S protein to ACE2 is significantly impaired in CFBE41o- (null) cells compared to cells over-expressing the F508del- (192%) and wild-type-CFTR (305%) (Fig. 7a, b).

In addition, we evaluated the effect of ACE2 ligand SARS-CoV-2 S protein on IL-6 expression. CF lung pathology has been established as a proinflammatory condition featuring elevated levels of cytokines and chemokines, in particular IL-8[37–39]. Our data consistently showed that

IL-6 mRNA expression was constitutively reduced in primary nasal and bronchial epithelial cells obtained from CF patients compared to healthy donor-derived tissues (Fig. 7c). Interestingly, knockdown of *CFTR* expression in SH3 cells reduced expression of *IL6* mRNA compared to mock-transfected Alter cells (Fig. 7c). Most importantly, stimulation with the SARS-CoV-2 S protein led to a remarkable induction of IL-6 release (14-fold increase) in primary hBEC grown at the air-liquid interface derived from healthy donors while induced a very weak response in CF primary cells (Fig. 7d). Importantly, similar results were obtained in hNEC (Fig. 7e). Although the recombinant SARS-CoV-2 S protein we employed has been certified as endotoxin-free, we nevertheless checked the effect of anti-S neutralizing antibody to evaluate whether the pro-inflammatory effect was induced specifically by the S protein. As reported in Fig. 7d, pre-incubation of S protein with neutralizing antibodies led to a significant reduction of IL-6 release upon the stimulation sustained by the immunocomplex compared to S protein alone in hBEC.

## Discussion

This work provides further insights into the regulation of expression and localization of ACE2 receptor in relation to CFTR channel expression and function. This issue is of substantial interest for unraveling additional aspects in the pathogenesis of SARS-CoV-2 infection. Here, we show that ACE2 expression and subcellular localization are clearly correlated with wild type or mutated CFTR protein. In fact, data emerged from CFBE41o- cells overexpressing the F508del-mutant or the wild-type *CFTR*, grown under polarized conditions, showed increased ACE2 protein levels. Being the F508del-mutated CFTR channel nonfunctional and mainly retained into the ER, our data indicate that CFTR function is not involved in regulation of ACE2 expression. Consistently with this finding, the functional inhibition of wild-type CFTR channel, sustained by the well-established inhibitor CFTR(inh)-172, did not promote any effects on ACE2 protein levels in bronchial epithelial cells grown under polarized condition, nor in primary hBEC. Our results showed also that different bronchial epithelial models expressing the F508del-mutant CFTR channel displayed mislocalized ACE2, which was retained into the ER together with CFTR. Finally, immunofluorescence analyses revealed that ACE2 and CFTR are generally co-localized in bronchial epithelial cells.

The correlation between CFTR expression and ACE2 is even more evident in 16HBE14o- cells homozygous for the severe nonsense mutations W1282X- or G542X-*CFTR*, thus completely lacking expression of the chloride channel. In these models, ACE2 is poorly expressed. To confirm this result, we evaluated ACE2 expression in a primary CF bronchial epithelium obtained from a patient harboring two class I mutations in *CFTR* gene, namely W1282X and 2184insA[40]. As expected, we found a remarkably reduced expression of ACE2 in this patient, even compared with F508del patients.

We then sought to verify if the reduced ACE2 expression observed in CF cells is associated with reduced SARS-CoV-2 recognition. Using different cell models in which CFTR expression was knocked-out by gene editing (i.e., cells expressing W1282X- and G542X-*CFTR*) or by transient silencing, using a novel miR-145-5p approach, we observed that loss of CFTR is associated with significant reduction in SARS-CoV-2 entry and replication. In line with our data, it has been recently reported that SARS-CoV-2 replication is reduced in hBEC obtained from F508del-*CFTR* homozygous patients[41].

Of note, SARS-CoV-2 S protein had higher binding affinity for human ACE2 compared to S protein derived from SARS-CoV(-1), thus possibly justifying why SARS-CoV-2 is more infectious, displaying a replication number over 3-fold higher than SARS-CoV(-1)[42]. Data from proximity ligation assay showed that parental CF bronchial epithelial CFBE41o- (null) cells displayed a remarkably reduced recognition of SARS-CoV-2 S protein, whereas the over-expression of wild-type *CFTR* in the same cell model promoted significant increase of SARS-CoV-2 S

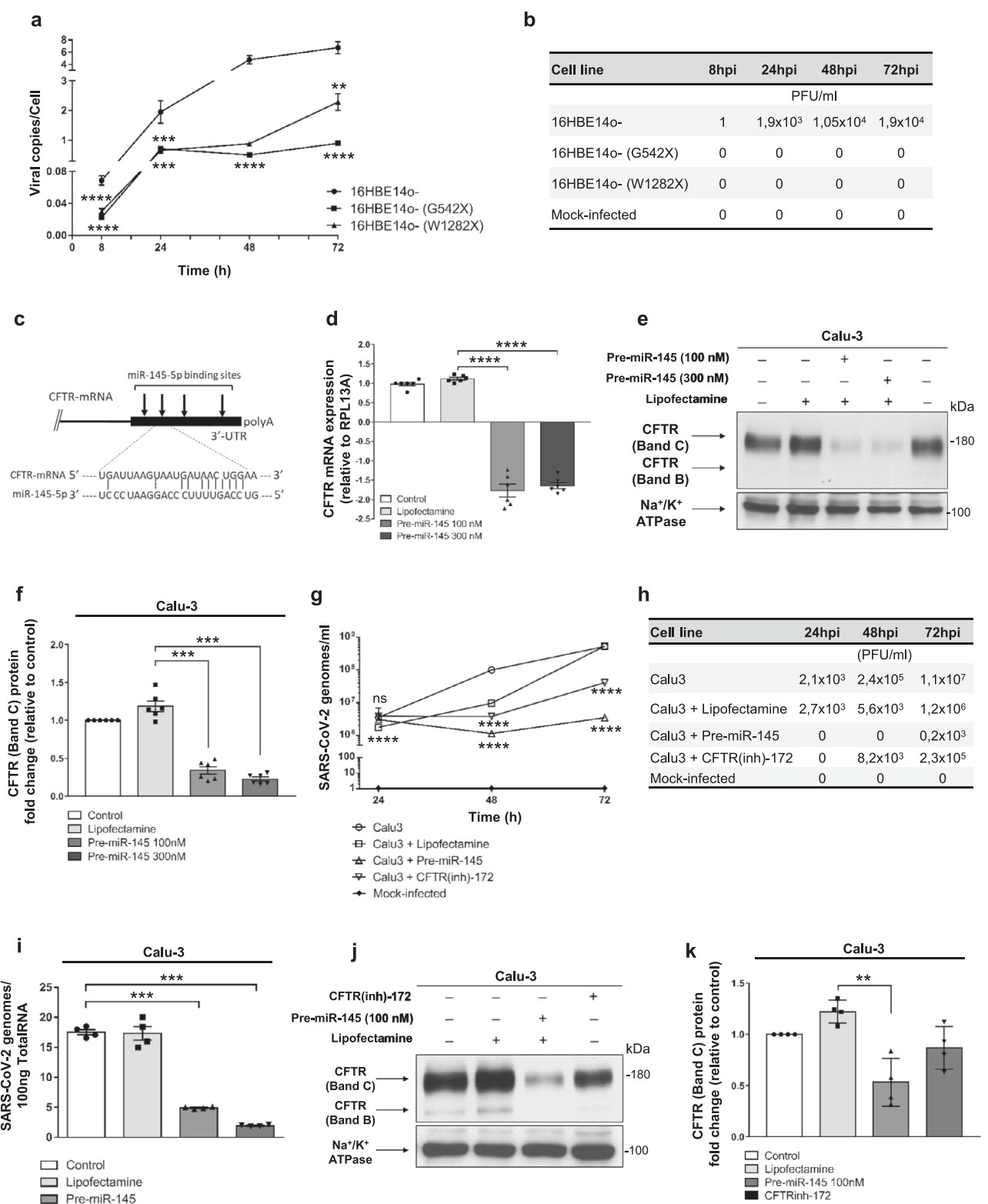

The binding of SARS-CoV-2 S protein to ACE2 receptors expressed
on the surface of host cells stimulates the release of IL-6 and other pro-
inflammatory mediators, thus fostering CRS[4,5]. It has been observed
that this process is mainly driven by NF-kB activation in different cell
models[43,44]. Most importantly, it has been recently reported that pri-
mary macrophages isolated from CF patients release lower levels of IL-
6 compared to healthy donor-derived cells upon stimulation with
SARS-CoV-2 S protein[45]. Thus, we sought to evaluate whether the

protein binding. Once again, therefore, overexpression of nonfunc-
tional F508del-*CFTR* seems to be sufficient to upregulate ACE2 and
promote S protein recognition, although this process is reduced by
37% compared to wild type *CFTR* overexpressing model. The hypoth-
esis is that over-expression of F508del-*CFTR* induces an increase of
ACE2 expression level, which in part enriches the plasma membrane
while the remainder amount of protein is retained within the endo-
cellular quality control mechanisms together with mutant CFTR.

**Fig. 6 | Loss of CFTR expression and function inhibits SARS-CoV-2 entry and replication. a** 16HBE14o- and mutant clones carrying W1282X- and G542X-CFTR were infected with 0.1 MOI SARS-CoV-2 for 1 h, and then the viral copies/cell in cell lysates was evaluated 8, 24, 48, and 72 hpi by dPCR. The results are shown as the mean ± SEM from three independent experiments (*n* = 3). **b** Viral titration was performed on supernatants of 16HBE14o- cells and their mutant clones infected with 0.1 MOI SARS-CoV-2 or 0.1 moi UV-inactivated SARS-CoV-2 (mock) for 1 h, and then the viral titration was performed by plaque assay after 8, 24, 48, and 72 hpi. **c** Location of the miR-145-5p binding sites within the CFTR 3'-UTR (binding site displaying the highest affinity is shown). **d–f** Effect of premiR-145-5p treatment on *CFTR* expression in Calu-3 cells. The effect was verified by RT−qPCR (**d**) and western blot analysis (**e**) in six independent experiments (*n* = 6). **f** Densitometry analysis of six independent experiments (*n* = 6), conducted as reported in (**e**). Data represented in panels **d–f** are shown as the mean ± SEM. Effect of premiR-145-5p compared to CFTR(inh)−172 on the extracellular release of SARS-CoV-2 in infected Calu-3 cells. Extracellular release of SARS-CoV-2 genomes (**g**); viral titration has been performed in Calu-3 cells upon SARS-CoV-2 infection sustained for 24, 48, and 72 hpi (**h**); intracellular production of SARS-CoV-2 genomes (**i**). Results are reported as the mean ± SEM from four independent experiments (*n* = 4). Western blot analysis of CFTR (**j**) and corresponding densitometry analysis (**k**) in response to SARS-CoV-2 infection under the same conditions depicted in panels (**g–i**). Results are reported as the mean ± SEM from four independent experiments (*n* = 4). Normal distribution was tested by the Shapiro−Wilk test before running the two-tailed Student's t test for paired data (**a, d, f, g, i, k**), which has been reported within the scatter plot with bars (**p < 0.01; ***p < 0.001; ****p < 0.0001). In panels **a** and **g**, for some points, the error bars were shorter than the height of the symbols. In this case, the error bars were not drawn. Source data are provided as a Source Data File.

reduced recognition of SARS-CoV-2 S protein, caused by partial loss of ACE2 expression and mislocalization observed in CF cells, was associated with decreased IL-6 release in primary airway epithelia. Our results suggest that CF condition may hamper the CRS triggered by SARS-CoV-2 S protein stimulation in airway epithelia, thus strengthening the hypothesis that CF may constitute a biological advantage by decreasing the risk of developing unfavorable COVID-19 outcomes.

CFTR stabilization on plasma membrane is facilitated by a multiple protein complex, in which NHERF1 plays a substantial role[30,46]. Recently, it has been reported that NHERF1 can directly interact with ACE2 receptor, through the PDZ-binding motif, facilitating SARS-CoV-2 recognition[25]. Although these data strengthen the hypothesis of a close interdependent association among ACE2, CFTR and NHERF1, our results show that CFBE41o- (null) cells, expressing undetectable levels of CFTR[29] and ACE2, displayed normal subcellular localization and expression of NHERF1 instead.

Despite ACE2 expression being unaffected by CFTR functional inhibition, we observed that blockage of CFTR ion efflux by CFTR(inh)-172 compound[26] leads to reduced SARS-CoV-2 viral infectivity and replication. These data suggest that CFTR functional inhibition might also influence the packaging process of viral replication, resulting in defective virions. This is not surprising, since a recent report explained that another chloride channel (i.e., TMEM16F) is involved in SARS-CoV-2 recognition and propagation among the airway epithelia[19]. Similarly to what we observed for CFTR, functional inhibition of TMEM16F was sufficient to reduce SARS-CoV-2 infectivity and syncytia formation[19]. Our hypothesis is that the CFTR channel might play a dual role in the regulation of SARS-CoV-2 life cycle. On the one hand, defective *CFTR* expression downregulates *ACE2* expression, thus negatively affecting SARS-CoV-2 viral entry. On the other hand, chloride channels, including CFTR, are important for SARS-CoV-2 biogenesis inside the host cells, mostly affecting viral replication. Overall, this study clarifies why CF condition, despite being potentially considered a comorbidity for COVID-19, is not associated with particularly severe outcome upon SARS-CoV-2 infection. Furthermore, this work suggests that CFTR and possibly other chloride channels could be taken into consideration as molecular targets for the development of alternative anti-COVID-19 therapies.

## Methods

### Human samples
All human samples were obtained and analyzed in accordance with the Declaration of Helsinki after written consent was obtained. All protocols were approved by the Ethics Committee of the Azienda Ospedaliera Universitaria Integrata (Verona, Italy), approval No. 2917CESC. Well-differentiated Mucil-air® primary human nasal and bronchial epithelial cells from F508del/F508del and 2184insA/W1282X CF patients (CF-hNEC) or from healthy donors (HNEC) were supplied by Epithelix (Plan-les-Ouates, Switzerland) after strict quality control was performed by the supplier. HNECs were cultured on Snapwell supports with Mucil-air® differentiating medium (Epithelix, Plan-les-Ouates, Switzerland) as previously described[47]. In addition, further cell lysates from primary human bronchial epithelial cells from F508del/F508del and 2184insA/W1282X CF patients (CF-hBEC) or from healthy donors (hBEC) were supplied by Epithelix. The human samples analyzed in this study are summarized in supplementary Table 1.

### Cell culture
Immortalized bronchial epithelial cells 16HBE14o- cells expressing wild-type CFTR, and isogenic cell lines, gene edited by CRISPR/Cas9 technology, expressing W1282X- and G542X-*CFTR* were supplied by the Cystic Fibrosis Foundation Therapeutics Lab (CFFT, Lexington, MA). CF bronchial epithelial CFBE41o- cells with or without stable expression of F508del-CFTR or wild-type CFTR obtained by Dr. J.P. Clancy[48] cells were cultured in MEM supplemented with 10% fetal bovine serum (FBS) and 2 mM L-glutamine in the absence of antibiotics. The submucosal gland cell line Calu-3, generated from bronchial adenocarcinoma, together with SH3 and Alter cells were kindly provided by Dr M. Chanson[27]. Briefly, stable expression of short hairpin RNAs (shRNAs) against CFTR was induced in Calu-3 cells by transfecting the Sleeping Beauty transposon vector pT2/si-PuroV2, generating a CFTR knockout cell line (SH3). A scrambled shRNA sequence was used to generate a mock transfected cell line (Alter), as previously described[27]. Calu-3, SH3, and Alter cells were maintained in DMEM/F12 (3:1 vol/vol) supplemented with 10% FBS without streptomycin or penicillin but were continuously selected in the presence of 4 µg/ml puromycin.

For cell polarization, transwell inserts (BD Biosciences, Franklin Lakes, NJ) were coated with a solution containing 10% human fibronectin (Sigma Aldrich, St. Louis, MO) (Thermo Fisher Scientific, Waltham, MA), 1% Bovine collagen (Thermo Fisher Scientific, Waltham, MA), 0.001% 1 mg/ml Bovine serum albumin (Sigma Aldrich, St. Louis, MO) in LHC-Basal medium. $5 \times 10^5$ cells were seeded in 200 µl Serum-free DMEM (Life Technologies, Thermo Fisher Scientific, Waltham, MA) and Serum-free Ham's F12 (Life Technologies, Thermo Fisher Scientific, Waltham, MA) 1:1, supplemented with 5% FBS (Sigma Aldrich, St. Louis, MO). 700 µl of the same medium were added to the basolateral part of the insert. After 24 h, the medium was replaced in both the apical and basolateral parts of the inserts with Serum-free DMEM (Life Technologies, Thermo Fisher Scientific, Waltham, MA) and Serum-free Ham's F12 (Life Technologies, Thermo Fisher Scientific, Waltham, MA) 1:1, supplemented with Ultroser G Serum Substitute (Sartorius, Goettingen, Germany). After one week, the Trans-Epithelial Electrical Resistance (TEER) was measured using Evom voltmeter (World Precision Instruments, Sarasota, FL). Polarization was reached when cells exhibited a >500 Ω resistance.

**Inhibition of CFTR expression by pre-miR-145-5p.** Transfection procedure of pre-miR-145-5p in Calu-3 cells was performed in 12-well plates using Lipofectamine RNAiMAX Transfection Reagent

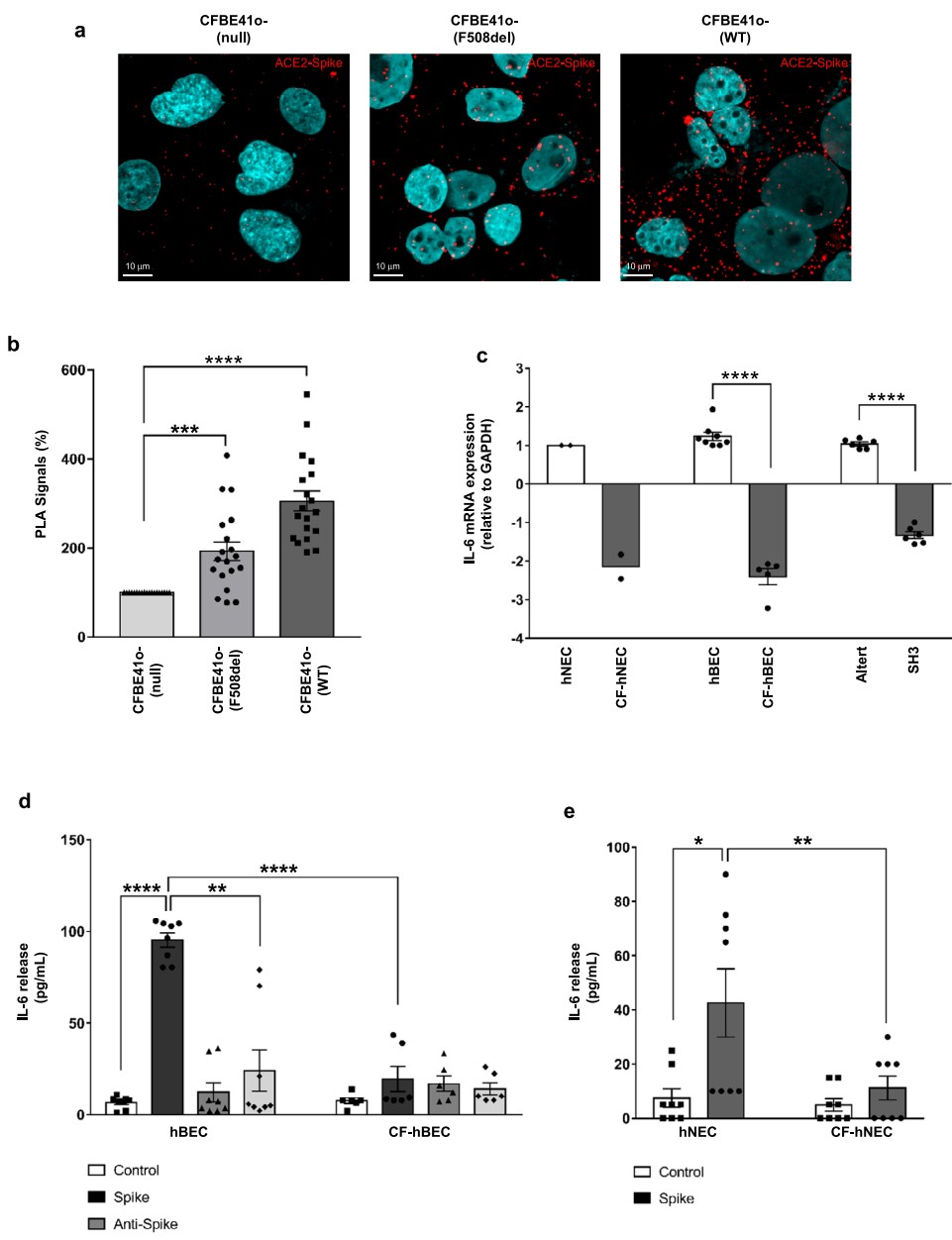

**Fig. 7 | SARS-CoV-2 Spike binding to ACE2 is decreased in bronchial epithelial cells lacking CFTR expression and IL-6 release induced by S protein is reduced in CF. a** Proximity ligation assay (PLA) for SARS-CoV-2 S protein and ACE2 interactions was performed in CFBE41o-, CFBE41o- (F508del) or CFBE41o- (WT). Representative images with PLA signals obtained from the binding of S protein to ACE2 (red) in different cells are shown. The cell nuclei were stained with DAPI (blue). **b** The scatter plot with bars shows quantification of PLA signals (%), with respect to CFBE41o- (null) cells ($n = 19$ independent visual field for each condition of three independent experiments). **c** IL-6 mRNA was quantified by RT–qPCR in lysates obtained from differentiated hNEC cells obtained from a pool of fourteen healthy donors (hNEC, pool was indicated by ♦) versus two CF patients (CF-hNEC) homozygous for F508del *CFTR* or in primary hBEC obtained from healthy control subjects ($n = 8$) versus CF patients homozygous for F508del *CFTR* ($n = 5$), or in Alter cells ($n = 6$) versus SH3 cells ($n = 6$). Data represented in panels **b**, **c** are shown as the mean ± SEM. **d** Quantification of IL-6 release by ELISA in cell culture supernatants obtained from well-differentiated hBECs and CF-hBECs. Cells were stimulated on both apical and basolateral sides with 1 µg/ml recombinant SARS-CoV-2 S protein or with S protein neutralized by 3 µg/ml anti-SARS-CoV-2 Spike RBD antibody for 12 h. Data are reported as the mean ± SEM from four independent experiments performed in duplicate ($n = 8$). **e** IL-6 release quantification from hNEC and CF-hNEC stimulated with S protein. Data are reported as the mean ± SEM from four independent experiments performed in duplicate ($n = 8$). Normal distribution was tested by the Shapiro–Wilk test before running the two-tailed Student's t test (**b**–**e**), which has been reported within the scatter plot with bars (*$p < 0.05$; **$p < 0.01$; ***$p < 0.001$; ****$p < 0.0001$). Source data are provided as a Source Data File.

(Invitrogen, Thermo Fischer Scientific, Waltham, MA) accordingly to manufacturer's instruction. Calu-3 cells were seeded at $2.5 \times 10^5$/ml with 100 or 300 nM of hsa-miR-145-5p miRNA precursor (PM11480, Ambion, Thermo Fisher Scientific, Waltham, MA). After 72 h, cells were collected and total RNA was isolated using TRI Reagent™ (Sigma Aldrich, St. Louis, MO) and immediately converted to cDNA. For the

experiments with SARS-CoV-2 infection, cells were pre-treated with pre-miR-145-5p at 100 nM for 48 h before the infection.

## Quantitative PCR

Total RNA from HNECs, HBECs, Calu-3, SH3, Alter, CFBE41o- and 16HBE14o- cells was isolated using a High Pure RNA Isolation Kit

(Roche, Mannheim, Germany) following the manufacturer's protocol. RNA concentration was determined using a NanoDrop 2000 spectrophotometer (Thermo Fisher Scientific, Waltham, MA) and then stored at −80 °C until use. A total of 500 ng of RNA was reverse transcribed to cDNA using a High-Capacity cDNA Reverse Transcription Kit with random primers (Thermo Fisher Scientific, Waltham, MA) following the manufacturer's recommendations. A total of 25 ng of cDNA was used for each reaction to quantify the relative gene expression. cDNA was then amplified using qPCRBio SyGreen Mix (PCR Biosystems, Wayne, PA) and QuantiTect Primer Assays (Qiagen, Hilden, Germany) for ACE2 (Hs_ACE2_1_SG, NM_021804), IL-6 (Hs_IL6_1_SG, NM_000565) and GAPDH (HS_GAPDH_1_SG, NM_001256799). The analysis was performed using an AriaMx thermocycler (Agilent, Santa Clara, CA) and GeneAmp PCR system 9700 thermal cycler (Thermo Fisher Scientific, Waltham, MA). Changes in mRNA expression levels were calculated following normalization to the GAPDH reference gene, and relative quantification was performed using the comparative cycle threshold method.

In experiments employing the pre-miR-145-5p, CFTR expression was analyzed by RT-qPCR using 300 ng of total RNA, which were reverse transcribed using the Taq-Man Reverse Transcription PCR Kit and random hexamers (Applied Biosystems, Thermo Fischer Scientific, Waltham, MA) in a final reaction volume of 50 μl. CFTR gene-specific double fluorescently labeled probe and primers were used (Assay ID: Hs00357011_m1). The relative expression was calculated using the comparative cycle threshold method and, as reference gene, the human RPL13A (Assay ID: Hs03043885_g1). Assays were purchased from Applied Biosystems. Data analyses were performed by QiaQuant 96 v.1.0.3 software (Qiagen, Hilden, Germany).

### Reverse Transcription droplet digital PCR

The number of copies of intracellular viral Spike sequence was assessed by Reverse Transcription droplet digital PCR (RT-ddPCR). Total RNA extracted (500 ng) from 16HBE14o-, G542X- and W1282X-*CFTR* cells at 8, 24, 48, and 72 hpi, were reverse transcribed using the Taq-Man™ Reverse Transcription PCR Kit with random hexamers (Thermo Fisher Scientific, Waltham, MA), according to manufacturer's manual, in a final reaction volume of 50 μl.

Nine μL or 1 μl of undiluted cDNA, form 8, 24, 48, and 72 hpi respectively, were amplified in a final volume of 20 μL ddPCR reaction, in the presence of ddPCR Supermix for Probes (no dUTP) 2X (Bio-Rad, Hercules, CA) and TaqMan 2019nCoV assay kit v1 RealTime-PCR (ThermoFisher, Waltham, MA) for viral Spike sequence. In parallel, in all the samples, RNAse P (TaqMan™ RNase P Control Reagents Kit, ThermoFisher, Waltham, MA) sequences were amplified as a loading control, using 1 μl of cDNA diluted 1:10.

Droplet emulsion was automatically generated using Automated Droplet Generator (AutoDG) (Bio-Rad, Hercules, CA) and amplified in GeneAmp PCR System 9700 thermal cycle (Thermo Fisher Scientific, Waltham, MA). The following thermal cycler conditions were used: 95 °C for 10 min, 40 cycles of 95 °C for 15 s and 60 °C for 1 min and a final step of 98 °C for 10 min.

Droplets were analyzed using the QX200 Droplet Reader, and data analysis was performed with QuantaSoft version 1.7.4 (Bio-Rad, Hercules, CA).

### Western blot

Protein extracts were obtained from 16HBE14o- (and their mutants), CFBE41o- (and their overexpressing models), Calu-3, SH3, Alter and primary hNEC and hBEC by using RIPA buffer (Sigma Aldrich, St. Louis, MO, USA) additioned with cOmplete™ Protease Inhibitor Cocktail (Roche, Mannheim, Germany) and DTT (dithiothreitol) (Thermo Fisher Scientific, Waltham, MA). Protein samples were quantified using the Quick Start™ Bradford Protein Assay (Bio-Rad Laboratories, Hercules, CA).

A total of 40 μg of cell extracts were denatured for 5 min at 95 °C in Laemmli Sample Buffer (Bio-Rad Laboratories, Hercules, CA), containing 355 mM 2-mercaptoethanol. For ACE2 and β-actin analysis, protein extracts were loaded on Miniprotean TGX (4–15%) SDS–PAGE gel (Bio-Rad Laboratories, Hercules, CA) in Tris-glycine buffer (25 mM Tris, 192 mM glycine, and 0.1% SDS) using Precision Plus dual color tag protein ladder (Bio-Rad Laboratories, Hercules, CA) to determine molecular weight. Membranes were probed with primary anti-human ACE2 rabbit IgG polyclonal antibody (ab15348, Abcam, Cambridge, UK, dilution 1:1000), and then with mouse anti-rabbit IgG-horseradish peroxidase-coupled secondary antibody (cod. #7074 s, Cell Signaling, Danvers, MA, dilution1:10,000). For CFTR analysis, 20–40 μg of the total protein extracts were heated in XT buffer (Bio-Rad Laboratories, Hercules, CA) at 37 °C for 10 min and loaded onto a 3–8% Tris-acetate gel (Bio-Rad Laboratories, Hercules, CA, USA). Proteins were transferred to polyvinylidene difluoride (PVDF) membranes (Bio-Rad Laboratories, Hercules, CA, USA) using Trans Blot Turbo (Bio-Rad Laboratories, Hercules, CA) and processed for western blotting using a mouse IgG monoclonal antibody against the NBD2 domain of CFTR (596, cod. A4, University of North Carolina, Chapel Hill, NC) at a dilution of 1:2500 with overnight incubation at 4 °C. After washing, the membranes were incubated with horseradish peroxidase-conjugated goat anti-mouse IgG (cod. 115-035-062, Jackson Immunoresearch, Cambridge, UK, dilution 1:10000) at room temperature for 1 h, and the signal was developed by enhanced chemiluminescence (LumiGlo Reagent and Peroxide, Cell Signaling, Denver, CO). After membrane stripping, a HRP-conjugated mouse monoclonal anti-β-actin antibody (clone AC-15, cod. A3854, Sigma Aldrich, St. Louis, MO, dilution 1:10000) was used to ensure equal loading of samples. For experiments employing the pre-miR-145-5p, cellular extracts were sonicated for 3 × 30 s on ice at 50% amplitude using the Vibra-Cell VC130 Ultrasonic Processor (Sonics). The mouse monoclonal anti-Na$^+$/K$^+$ ATPase IgG antibody (SC-514614, Santa Cruz Biotechnology, Santa Cruz, CA, dilution 1:500) was used as housekeeping (loading control). Data analysis was performed by ImageLab Touch v. 3.0.1.14 (Bio-Rad, Hercules, CA) and ImageJ v.1.8.0_172 (NIH, Bethesda, MD).

### Sectioning of airway epithelia and hematoxylin/eosin staining.

Inserts containing airway epithelia were fixed in 4% paraformaldehyde and put in labeled embedding cassettes. Increasing concentrations of ethanol (70, 96, and 100%) and xylene were used to obtain sample dehydration. The cassettes were then transferred to the melted paraffin for infiltration, and heated for two hours. Then samples were transferred to the mould and covered with paraffin. Samples were sectioned using the automated rotary microtome HistoCore Autocut (Leica Biosystems, Wetzlar, Germany). Sections were cut at a thickness of about 4–5 μm and flattened out by floating on surface of heated (40–50 °C) ultrapure water. Microscope slides were used to pick the sections before drying for two hours at 60 °C.

Hematoxylin and Eosin (H/E) staining was performed by H&E Stain Kit (Abcam, Cambridge, UK) following the manufacturer's instructions.

### Immunofluorescence

Cells were seeded onto 24 mm coverslips. The next day, the cells were rinsed with ice-cold PBS and fixed in 4% paraformaldehyde for 20 min at room temperature. To eliminate paraformaldehyde autofluorescence, cells were incubated for 10 min at room temperature with a 0.1 M glycine solution, washed and permeabilized with 0.1% Triton X-100. Nonspecific binding of antibodies was prevented by incubating cells with a 2% BSA solution for an hour at room temperature. For polarized airway epithelia assays, slides containing paraffin embedded sections derived from polarized cells or hBEC grown in transwell inserts were employed.

Subsequently, a double immunofluorescence procedure was performed, and the cells were incubated overnight at 4 °C with a

mouse monoclonal IgG antibody against CFTR R-domain (570, cod. A2 from University of North Carolina, Chapel Hill, NC, 1:200) and a rabbit polyclonal anti-ACE2 IgG (ab15348, AbCam, Cambridge, UK, dilution 1:200), or in a sequential procedure, with mouse anti-CFTR IgG (570, University of North Carolina, Chapel Hill, NC, 1:200) and mouse IgG anti-NHERF1 (611161 from BD Transduction Laboratories, 1:200). Cells were then washed three times with cold PBS and incubated for 1 h at room temperature with goat anti-mouse Alexa-488 (A32723, Thermo Fisher Scientific, Waltham, MA, dilution 1:1000) and goat anti-rabbit Alexa-594 (A11012, Thermo Fisher Scientific, Waltham, MA, 1:1000) or, in the second case, with goat anti-mouse Alexa-488 (cod. A32723, Thermo Fisher Scientific, Waltham, MA, dilution 1:1000) and rabbit anti-mouse Alexa-594 (A27027, Thermo Fisher Scientific, Waltham, MA, 1:1000) antibodies. Cells were then mounted on a coverslip with Pro-Long Gold antifade reagent (Invitrogen, Thermo Fisher Scientific, Waltham, MA) and examined by Zeiss LSM510 confocal fluorescence microscopy. The degree of colocalization between CFTR channel and ACE2 receptor was quantified using the Pearson correlation coefficient and the Mander's overlap coefficient[49] by Zeiss Zen 2009 software v.6.0.0.303.

### Proximity ligation assay

Proximity ligation assay to quantify SARS-CoV-2 S protein and ACE2 receptor interactions was performed according to the manufacturer's protocol of Duolink reagents (Sigma-Aldrich, St. Louis, MO, USA). Cells were fixed in 4% paraformaldehyde in KRB and probed with mouse monoclonal anti-ACE2 (E-11) IgG (cod. SC-390851 Santa Cruz Biotechnology, Santa Cruz, CA, dilution 1:200) and rabbit anti-SARS-CoV-2 S protein IgG (cod. NBP3-11940 Novus Biologicals, Bio-Techne SRL, Milan, Italy, dilution 1:200) antibodies. Signals were developed using a Duolink In Situ Far Red kit. Images were acquired with confocal laser scanning Olympus FV3000 microscope equipped with Fluoview FV software v.31s-sw, oil 63× objective and quantified using ImageJ software (NIH, Bethesda, MD).

### SARS-CoV-2 propagation and infection

SARS-CoV-2 was isolated from a nasopharyngeal swab retrieved from a patient with COVID-19 (Caucasian man of Italian origin, genome sequences available at GenBank (SARS-CoV-2-UNIBS-AP66: ERR4145453)). This SARS-CoV-2 isolate clustered in the B1 clade, similar to most Italian sequences, together with sequences derived from other European countries and the United States. SARS-CoV-2 inoculum (a kind gift of Professor Arnaldo Caruso, University of Brescia) was obtained in VeroE6 cells (ATCC, Manassas, VA, Number CRL-1586). VeroE6 and Calu-3 (ATCC, Manassas, VA, HTB-55) cell lines were cultivated and maintained in Modified Eagle Medium (MEM; Gibco, Waltham, MA) supplemented with 10% heat-inactivated fetal calf serum (FCS) at 37 °C in a humidified atmosphere of 5% $CO_2$. For the infection protocol, FCS was decreased to 2%. Vero E6 were infected with serial dilutions of the cell supernatants collected at different time points post-infection. After 1 h virus absorption, complete medium with 2% methylcellulose was added. Five days after infection, cells were methanol-fixed, and plaques were stained with crystal violet (0.1%) and counted. The experiments were performed in triplicate[31]. SARS-CoV-2 manipulation was performed in the BSL-3 laboratory of the University of Ferrara, following the biosafety requirements. Calu-3, 16HBE14o- and their mutants cell susceptibility to SARS-CoV-2 infection was assayed by infecting single cells with a multiplicity of infection (MOI) of 0.1 for 2 h at 37 °C, as previously reported (approx. $2 \times 10^5$ infectious virus particles per well). We used UV-irradiated SARS-CoV-2 virions as mock control. Briefly, SARS-CoV-2 virions were treated for 20 min at 15 cm distance under UV irradiation (1350 μW × 20 min × 60 s = 1620 mJ/cm$^2$ at 365 nm). Eight, 24, 48 and 72 h after infection, the infected cells were collected.

### Viral RNA detection

RNA extraction was performed 8, 24, 48, and 72 h post infection (hpi) using a MagMAX Viral/Pathogen Nucleic Acid Isolation kit (Thermo Fisher, Waltham, MA) for recovery of RNA and DNA from the virus, as previously described[31]. SARS-CoV-2 titration was obtained using a TaqMan 2019nCoV assay kit v1 Real Time-PCR (Thermo Fisher, Waltham, MA).

### Viral titration

Titer determination of infective SARS-CoV-2 virions was performed by plaque assay on VeroE6 cells (ATCC, Manassas, VA, Number CRL-1586). VeroE6 were infected with serial dilutions of the cell supernatants collected at different time points post infection. After 1 h virus absorption, the complete medium with 2% methylcellulose was added. Five days after infection, cells were methanol-fixed, and plaques were stained with crystal violet (0.1%) and counted[31]. The experiments were performed in triplicate.

### Cytokine assays

Primary hBECs grown in air-liquid interface were incubated for 12 h with the following treatments: 1 μg/ml SARS-CoV-2 spike protein (Sino Biological, Chesterbrook, PA); 3 μg/ml anti-SARS-CoV-2 Spike RBD antibody (ab281303, Abcam, Cambridge, UK); 1 μg/ml SARS-CoV-2 spike neutralized by 3 μg/ml anti-SARS-CoV-2 Spike RBD antibody. The Spike immunocomplex was obtained by incubating S protein and ant-Spike RBD antibody overnight at 4 °C. Stimuli were added both to the apical side (60 μl) and basolateral side (700 μl). Cell supernatants were then collected for cytokine assays. IL-6 released into cell culture supernatants was measured by ELISA (ab46027, Abcam, Cambridge, UK) following the manufacturer's protocol. Samples were subsequently analyzed on a Sunrise microplate reader (Tecan Trading AG, Männedorf, Switzerland).

### Statistical analysis

Shapiro–Wilk test was used to evaluate normal distribution in each experiment. According to this evaluation, independent group determination was tested using two-tailed Student's t-test for paired or unpaired data. A $p$-value $<0.05$ was considered statistically significant. The statistical software Prism 7 (GraphPad, San Diego, CA) was used.

### Reporting summary

Further information on research design is available in the Nature Portfolio Reporting Summary linked to this article.

## Data availability

The raw data generated in this study are provided in the Source Data file. Source data are provided with this paper.

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

## Acknowledgements

We are grateful to the Cystic Fibrosis Foundation (Bethesda, MD) for kindly providing the CRISPR/Cas9 gene edited 16HBE14o-, W1282X-*CFTR* and G542X-*CFTR* cell lines, to Dr. J.R. Riordan (University of North Carolina, Chapel Hill) for anti-CFTR ab596 and ab570 (Cystic Fibrosis Foundation), to Alessandro Sorio, Manuela Beverari and Federica Quiri (Azienda Ospedaliera Universitaria Integrata, Verona, Italy) for the excellent technical support, and to Marc Chanson (University of Geneva) for providing Calu-3 SH3 and Alter cell lines. A.R. is supported by local funds from University of Ferrara, FIR-2021, the Italian Ministry of Health (GR-2016-02364602) and the Italian Ministry of Education, University and Research (PRIN Grant 2017XA5J5N). P.P. is supported by the Italian Association for Cancer Research (AIRC, IG-23670), Telethon (GGP11139B), local funds from the University of Ferrara, and the Italian Ministry of Education, University and Research (PRIN Grant 2017E5L5P3). AF and RG are funded by MUR-FISR COVID-miRNAPNA (Project FIS-R2020IP_04128). S.V. and M.C. are funded by Pfizer Cybergrants, grant number ID61509709. This work was partially funded by the following (alphabetically): Chiesi Farmaceutici (Parma, Italy), Mylan N.V. (Hatfield, UK). This work is part of the CF Italian Task Force Against COVID-19 Action (CF-ITACA) of the Italian Cystic Fibrosis Society (SIFC), which partially supported this work as well.

## Author contributions

V.B. conceived the idea, co-designed the study, analyzed the data, and wrote the manuscript; V.G. performed the in vitro viral infections and analyzed the data; M.A., C.B., E.B., D.O., M.D., and E.T. performed experiments concerning ACE2 and CFTR expression and ELISA; A.F. and C.P. performed the experiments with pre-miR-145-5p and RT-ddPCR for viral copy quantification on infected polarized cells; SL performed the IF experiments; A.T., S.V., and G.C. analyzed the data and internally revised the manuscript; M.B. designed and performed the experiments on cytokine expression; P.P., F.B., and G.L. critically reviewed the final manuscript and gave the final approval of the version to be published; R.G. designed the pre-miR-145-5p experiments and internally reviewed the manuscript; A.R. designed the IF and PLA experiments, analyzed the data, and internally reviewed the manuscript; R.R. designed and supervised the experiments concerning in vitro viral infections; M.C. conceived the idea, co-designed and supervised the study.

## Competing interests

The authors declare no competing interests.
