## [Peer Review File · Nature Communications]

SARS-CoV-2 viral entry and replication is impaired in Cystic Fibrosis airways due to ACE2 downregulationREVIEWER COMMENTS

Reviewer #1 (Remarks to the Author):

The fact that several studies have reported that CF seems to protect against COVID-19, warrants further research. Bezerrie et al., now report that ACE2 expression and localization are regulated by cystic fibrosis transmembrane conductance regulator (CFTR) channels. Overall there are quite a number of experiments analyzing the regulation of ACE2 expression in different cell lines. However, the analysis of susceptibility of cells with reduced ACE2 expression due to this defect to infection with SARS-CoV-2 is rather limited. In addition, confirmation of the reduced ACE-2 expression needs to be confirmed by analysis of spike protein binding on these cells.

Given the fact that diminished ACE2 expression would be important in the very early processes involved in the life cycle of SARS-CoV-2, viral entry, more focus should be on this aspect.

First, as a result of the diminished ACE-2 expression binding of the virus is expected to be affected. This could be easily demonstrated by using recombinant spike protein binding to cells used.

In addition, more focus should be on the early kinetics of viral replication in the different cells. Is the nr of infected cells reduced? This can be checked at for instance 8-10 h after infection by a staining of SARS-CoV-2 infected cells. Now the growth curves show mainly later time points (24- 72 h post infection). Pseudotyped viruses could also be used for these experiments.

Importantly, standard deviation should be determined as in the case of figure 5a and 5f only single determinations are shown. A statistical analysis should be performed as well.

If the main effect is on entry, how do the authors explain that there is actually no difference in viral RNA at 24 h post infection? A more detailed analysis is really needed, if possible also with titrations of infectious virus in addition to the RNA quantifications. The Y-axis should be modified to include lower values (Fig 5a)

Further controls are needed in the experiment demonstrating IL-6 induction by the spike protein. Specificity of IL-6 induction by the spike protein needs to be shown. Potentially, contamination in this prep induced IL-6 (LPS?). SARS-CoV-2 blocking antibodies should inhibit the IL-6 induction....

Reviewer #2 (Remarks to the Author):

Epidemiological data suggesting people with CF are relatively protected from COVID-19 disease remains curious although both biological host factors and behavior factors (such as improved ability and motivation to self-isolate, a finding that should be stated here), remain quite possible. Thus, studies that evaluate the biological basis of these observations remains of interest, and this manuscript adds to the work in a novel way. However, the experiments conducted largely rely on cell line data, particularly the CFBE41o- line, which is known to alter ERAD, were performed under unpolarized conditions that do not allow CFTR to process efficiently, and because of their over expression transgene, substantially induce ERAD compared to native expression cells. This major factor, plus some inconsistencies regarding the role of CFTR inhibitor vs. CFTR mutation type, make interpretation confusing, dampening enthusiasm in its present form. There is also a lack of appropriate statistical presentation in several critical experiments that should be addressed. Critiques are presented below:

1. CFTR is highly overexpressed in CFBE41o- cells with the complementary WT or mutant CFTR transgenes. This influences ER stress and CFTR localization, especially when assessed in non-polarized formats. The studies in Figure 2 need to be performed under polarized conditions in a cell with endogenous CFTR expression, to prevent the risk that this is an artifact of the overexpression

system. The ideal cell type would be primary HBE, the gold standard for CFTR biogenesis, but 16HBE with or without CFTR mutations (F508del and null) would be a reasonable stand-in if grown under polarized conditions. This is particularly critical for the SARS-CoV-2 studies, where the same mutant cell lines that showed the abnormal cell biology should be examined— namely CFTR mutant cells with F508del and with a null allele, and under endogenous CFTR expression.

2. There is an inconsistency regarding whether the effects of CFTR and ACE2 are due to loss of expression, function, or both, and this incongruity occurs throughout the paper.
 - a. -Figure 1f,g suggests mutated non-functional F508del restores ACE2 expression, directly contradicting the conclusions that ACE2 'is more widely related to defective CFTR protein'
 - b. -Yet Figure 3 invokes that F508del should not restore ACE2 expression
 - c. -Figure 1h,i suggests CFTR inhibitor is sufficient to reduce ACE2 expression, a condition not tested in other cell lines for comparison
 - d. -But Figure 2 shows CFTR inhibitor did not influence ACE2 expression patterns

3. The statistical design and figure presentation are problematic in several important locations in the manuscript:
 - a. Figure 1a should show individual data points as they are derived from individual donors. The normalization does not look reliable, in that the distribution among normal donors must be larger than that shown for each of the factors shown if they were run in a grouped basis as expected. Figure 1c is not called out in text and has the same problem. Figure 1b: protein levels in more donors should be shown – if they were run only in duplex, then the normalization of normal may not be reliable and inflate statistical comparisons.
 - b. Figure 5a needs error bars and mock controls. Statistics are not presented for this key experiment on SARS-CoV2 replication. Same with 5f.
 - c. Coefficient studies should show the plots and distribution of individual replicates.

4. The effects of CFTR expression and function on CD13 also vary widely by cell type, and the logic for examining CD13 is not presented.

Minor Comments:

1. It should be noted that 16HBE14o- cells are functionally heterozygous due to the insertion of the SV40 sequence within one of the two CFTR sequences. This is misstated in the manuscript.
2. Reasons for the apparent regulatory effects of CFTR on other ion channels include its impact on total cell potential difference, a unifying mechanism for many of the ion channels reported.
3. Given the deleterious effects of inhibiting CFTR on lung health, the concept of inhibiting CFTR as a therapeutic strategy seems quite premature to introduce here. This is particularly true given the substantial effects of SARS-CoV-2 infection on respiratory epithelial, including deleterious effects on mucus clearance.

Ref. NCOMMS-21-44580A – Bezzetti et al.

Point-by-point response letter to the Reviewers

We thank the Editors and both the Reviewers for their helpful suggestions and comments. Following their recommendation, the revised manuscript has been remarkably improved, and the hypothesis has been consequently strengthened.

Our point-by-point responses are marked in red within the text.

Reviewer #1 (Remarks to the Author):

The fact that several studies have reported that CF seems to protect against COVID-19, warrants further research. Bezzetti et al., now report that ACE2 expression and localization are regulated by cystic fibrosis transmembrane conductance regulator (CFTR) channels. Overall, there are quite a number of experiments analyzing the regulation of ACE2 expression in different cell lines.

However, the analysis of susceptibility of cells with reduced ACE2 expression due to this defect to infection with SARS-CoV-2 is rather limited.

R1: According to the Reviewer comment, we expanded the analysis of susceptibility of cells to SARS-CoV-2 infection in other cellular models, including polarized 16HBE14o- (new Fig. 6a, b), besides expanding the previous results on polarized Calu-3 cells, which are still widely considered a golden standard for SARS-CoV-2 infections (new Fig. 6g-i). In addition, it should be noted that it has been recently reported that primary hBEC cells obtained from patients carrying F508del-CFTR mutations (the same primary cell model employed in our study) show reduced SARS-CoV-2 proliferation (Lotti et al, Cells, 2022). This reference has been added to the new version of the manuscript.

In addition, confirmation of the reduced ACE-2 expression needs to be confirmed by analysis of spike protein binding on these cells. Given the fact that diminished ACE2 expression would be important in the very early processes involved in the life cycle of SARS-CoV-2, viral entry, more focus should be on this aspect. First, as a result of the diminished ACE-2 expression binding of the virus is expected to be affected. This could be easily demonstrated by using recombinant spike protein binding to cells used. In addition, more focus should be on the early kinetics of viral replication in the different cells. Is the nr of infected cells reduced? This can be checked at for instance 8-10 h after infection by a staining of SARS-CoV-2 infected cells. Now the growth curves show mainly later time points (24- 72 h post infection). Pseudotyped viruses could also be used for these experiments.

R2: We thank the reviewer for the helpful suggestion. Following the Reviewer recommendation, we analyzed SARS-CoV-2 Spike binding to ACE2 using proximity ligation assay (PLA). Results confirmed that remarkably reduced levels of ACE2 observed in CFBE41o- (null) cells is associated with significantly impaired recognition of S protein by these cells. In addition, the re-introduction of CFTR expression led to increased binding of S protein to ACE2, as expected. These important findings are shown in new figure 7a, b.

In order to evaluate early kinetics of viral replication, we repeated the experiments evaluating the effect of SARS-CoV-2 infection also after 8 hours as suggested by the Reviewer, employing polarized 16HBE14o- cells and the gene-edited cell lines carrying the W1282X- and G542X-*CFTR* mutations. According to the Reviewers' recommendations we employed polarized 16HBE14o- cells and their mutants, in order to avoid altered ERAD-dependent experimental artifacts possibly resulting from CFBE41o- overexpressing *CFTR*. Digital PCR analysis (described in the amended method section, page 20-21) revealed that SARS-CoV-2 viral entry is significantly reduced in mutant cells starting from early time points (8h), as observed by digital PCR (new Fig. 6a) and by viral titration (plaque assays, new Fig. 6b).

Importantly, standard deviation should be determined as in the case of figure 5a and 5f only single determinations are shown. A statistical analysis should be performed as well.

R3: We agree with the Reviewer. We added SD and statistics, accordingly (please see new Fig. 6a and Fig. 6g). For some points, the error bars were shorter than the height of the symbols. In these cases, PRISM software did not draw the error bars. This point has been highlighted in the figure legend.

If the main effect is on entry, how do the authors explain that there is actually no difference in viral RNA at 24 h post infection? A more detailed analysis is really needed, if possible, also with titrations of infectious virus in addition to the RNA quantifications. The Y-axis should be modified to include lower values (Fig 5a)

R4: Viral titrations (plaque assay) have been added to new figure 6 (panels b and h), as suggested by the Reviewer. The Y-axis of the new Fig. 6a (polarized 16HBE14o- cells) has been expanded to include lower values, thus highlighting that a significant decrease in viral entry can be observed already at 8 and 24 hpi.

Further controls are needed in the experiment demonstrating IL-6 induction by the spike protein. Specificity of IL-6 induction by the spike protein needs to be shown. Potentially, contamination in this prep induced IL-6 (LPS?). SARS-CoV-2 blocking antibodies should inhibit the IL-6 induction....

R5: As declared by the manufacturer, the recombinant Spike protein employed was purified and endotoxin contamination was excluded by the LAL method (< 1.0 EU per µg protein). In order to check the specificity of IL-6 induction by the Spike protein, we pre-incubated Spike with a neutralizing anti-Spike antibody, as suggested by the Reviewer. The previous figure 5k has been replaced with the new figure 7d-e, showing that Spike protein induces high levels of IL-6 release in normal hNEC and hBEC, whereas neutralized Spike-dependent IL-6 levels are significantly reduced (new Fig. 7d). Moreover, our results confirm that S protein-dependent IL-6 release is hugely reduced in CF airway epithelia (both hNEC and hBEC, new Fig. 7d, e).

Reviewer #2 (Remarks to the Author):

Epidemiological data suggesting people with CF are relatively protected from COVID-19 disease remains curious although both biological host factors and behavior factors (such as improved ability and motivation to self-isolate, a finding that should be stated here), remain quite possible. Thus, studies that evaluate the biological basis of these observations remains of interest, and this manuscript adds to the work in a novel way.

R6: We thank the Reviewer for the kind appreciation in our work. According to the Reviewer's suggestion, we improved the description of the possible host and behavior factors protecting CF patients by COVID-19 (see lines 82-83).

However, the experiments conducted largely rely on cell line data, particularly the CFBE41o-line, which is known to alter ERAD, were performed under unpolarized conditions that do not allow CFTR to process efficiently, and because of their over expression transgene, substantially induce ERAD compared to native expression cells. This major factor, plus some inconsistencies regarding the role of CFTR inhibitor vs. CFTR mutation type, make interpretation confusing, dampening enthusiasm in its present form.

R7: According to the Reviewer recommendation, we repeated all the experiments using polarized cell models. In addition, it should be noted that together with over-expressing CFBE41o- (F508del and WT mutants) we employed the parental CFBE41o- cells, which constitutively exhibit undetectable levels of *CFTR* expression and has been considered therefore as null-*CFTR* cells (Sondo et al, Am J Physiol Cell Physiol 2011). Then, in order to avoid confounding results derived from increased ERAD, we employed different polarized cell models including the 16HBE14o- gene edited by the Cystic Fibrosis Foundation (Bethesda, MD) to express W1282X- and G542X-*CFTR*. Eventually, we re-tested key experiments on ALI-differentiated hBEC obtained from four different CF patients.

There is also a lack of appropriate statistical presentation in several critical experiments that should be addressed. Critiques are presented below:

1. CFTR is highly overexpressed in CFBE41o- cells with the complementary WT or mutant CFTR transgenes. This influences ER stress and CFTR localization, especially when assessed in non-polarized formats.

The studies in Figure 2 need to be performed under polarized conditions in a cell with endogenous CFTR expression, to prevent the risk that this is an artifact of the overexpression system. The ideal cell type would be primary HBE, the gold standard for CFTR biogenesis, but 16HBE with or without CFTR mutations (F508del and null) would be a reasonable stand-in if grown under polarized conditions.

R8: Following the Reviewer recommendations, we re-analyzed immunofluorescence assays in polarized CFBE41o- (null and overexpressing WT-*CFTR*) and 16HBE14o- (both parental and mutants) as well as in hBEC. Results on polarized 16HBE14o- are shown in new Fig. 4, which extends the previous Fig. 2. Both polarized 16HBE14o- cells and hBEC confirmed that ACE2 is poorly localized in the plasma membrane in polarized bronchial epithelial cells

carrying mutated *CFTR* (new Fig. 4). Since the previous Fig.2 depicting single-cell analysis was clearer in showing the ACE2 localization, we maintained that figure (now new Fig. 3).

This is particularly critical for the SARS-CoV-2 studies, where the same mutant cell lines that showed the abnormal cell biology should be examined– namely CFTR mutant cells with F508del and with a null allele, and under endogenous CFTR expression.

R9: According to the Reviewer suggestion, we evaluated SARS-CoV-2 infectivity in polarized 16HBE14o- (expressing wild-type *CFTR*) and in their mutant clones W1282X-*CFTR* and G542X-*CFTR*. This experiment has been performed also following the recommendation of another Reviewer and clarifies that SARS-CoV-2 infectivity is significantly reduced starting from early time points after infections (8 hours) (new Fig. 6a, b). Furthermore, it has been recently reported that primary hBEC cells obtained from patients carrying F508del-*CFTR* mutations (the same primary cell model employed in our study) show reduced SARS-CoV-2 proliferation (Lotti et al, Cells, 2022). This reference has been added to the new version of the manuscript.

2. There is an inconsistency regarding whether the effects of CFTR and ACE2 are due to loss of expression, function, or both, and this incongruency occurs throughout the paper.

a. -Figure 1f,g suggests mutated non-functional F508del restores ACE2 expression, directly contradicting the conclusions that ACE2 'is more widely related to defective CFTR protein'

R13: Although we observed a partial reduction of ACE2 expression (almost 25%) upon CFTR functional inhibition (using CFTRinh-172) in non-polarized 16HBE14o- cells, new results indicate that the functional inhibition of CFTR in polarized cell models, as well as in primary ALI-differentiated hBEC, is not able to reduce ACE2 expression. Thus, ACE2 expression seems to be mainly regulated by CFTR expression, not by CFTR function. On the contrary, CFTR function affects SARS-CoV-2 life cycle. The involvement of other chloride channels (i.e. TMEM16F) have been already recently reported as relevant contributors in SARS-CoV-2 infectivity (Braga et al, Nature, 2021). These new findings have been extensively discussed in the revised manuscript (lines 327-339). We thank the Reviewer for this helpful suggestion, which clarified the effect of CFTR functional inhibition in ACE2 expression.

b. -Yet Figure 3 invokes that F508del should not restore ACE2 expression

R14: Previous Fig. 3 (now moved in Supplementary Fig. 2), actually described the interaction between CFTR and NHERF1. Previous Fig. 2 only described the co-localization of ACE2 with CFTR instead, being not a quantitative assay. However, new Fig. 4 shows CFTR and ACE2 protein localization in polarized bronchial epithelial cells. In this figure, which is mainly aimed at describing the co-localization of ACE2 and CFTR in polarized bronchial epithelial cells, it is more evident that the loss of *CFTR* expression displayed by CFBE41o- (null) and 16HBE14o- cells carrying W1282X- and G542X-*CFTR* is associated with a dramatic reduction of ACE2 levels, mostly onto the apical plasma membranes of the epithelia. Similar results have been obtained in ALI-differentiated hBEC (new Fig. 5).

c. -Figure 1h,i suggests CFTR inhibitor is sufficient to reduce ACE2 expression, a condition not tested in other cell lines for comparison

R15: We agree with the Reviewer. Thus, we expanded the analysis on other polarized cell models and in primary hBEC. Results indicate that CFTR inhibitor is unable to modulate ACE2 expression (new Fig. 1d, f and Fig. 2d, e). This point has been clarified within the revised manuscript (lines 272-275).

d. -But Figure 2 shows CFTR inhibitor did not influence ACE2 expression patterns

R16: Previous figure 2 is mainly focused on ACE2 and CFTR co-localization, being not a quantitative assay, unfortunately. However, as stated above, we clarified that ACE2 expression is not influenced by the treatment with CFTR(inh)-172.

3. The statistical design and figure presentation are problematic in several important locations in the manuscript:

a. Figure 1a should show individual data points as they are derived from individual donors. The normalization does not look reliable, in that the distribution among normal donors must be larger than that shown for each of the factors shown if they were run in a grouped basis as expected.

R17: Figure 1a has been modified, showing individual data points, accordingly. Data shown in new Fig. 1a describe ACE2 mRNA expression in 2 hNEC from patients with CF, homozygous for F508del-*CFTR* mutation, compared with 2 healthy controls derived from a pool of 14 donors (highlighted by diamond plots instead of circle plots within the graph), as supplied by Epithelix. Each sample was analyzed in duplicate. This explains why distribution among normal donors is not large as one could expect.

In order to address the issue raised by the Reviewer, we analyzed ACE2 mRNA in six different hBEC isolated from CF patients homozygous for F508del-*CFTR* mutation, compared with six hBEC isolated from healthy individuals (as indicated in Supplemental Table 1). In this case, distribution is larger (especially in CF samples), as expected. Nevertheless, results obtained in hBEC confirmed that ACE2 mRNA is significantly downregulated in CF cells (new Fig. 1b).

Figure 1c is not called out in text and has the same problem.

Figure 1b: protein levels in more donors should be shown – if they were run only in duplex, then the normalization of normal may not be reliable and inflate statistical comparisons.

R18: We apologize for missing the figure legend of the previous Fig. 1c. According to the Reviewer recommendations, we deleted the statistical analysis from the previous Fig. 1a, c (ACE2 protein levels in hNEC), since data were derived from two samples and it was not reliable (new Fig. 1e). We therefore analyzed hBEC, expanding the sample size (N=6, Fig. 1b). Data from hBEC are consistent with those obtained from hNEC and show a significant reduction of ACE2 protein levels in CF cells (new Fig. 1d, f).

b. Figure 5a needs error bars and mock controls. Statistics are not presented for this key experiment on SARS-CoV2 replication. Same with 5f.

R19: According to this and another Reviewer, we modified the previous Fig. 5a with the new Fig. 6a, employing polarized cells which do not exhibit increased ERAD, namely 16HBE14o- and their mutants W1282X-CFTR and G542X-CFTR (kindly supplied by the Cystic Fibrosis Foundation, Bethesda, MD). New results indicate that SARS-CoV-2 infectivity is significantly reduced in CF polarized bronchial epithelial cells already starting from early time points (8 hours), consistently with the reduced expression of the receptor ACE2. Error bars and mock controls have been added. Error bars in new Fig. 6g (previous Fig. 5f) are smaller than the plot symbols, thus invisible (we added this specification within the figure legend).

c. Coefficient studies should show the plots and distribution of individual replicates.

R20: New Fig. 3 (previously reported as Fig. 2) and new Supplementary Fig. 2 (previous Fig. 3) have been modified accordingly.

4. The effects of CFTR expression and function on CD13 also vary widely by cell type, and the logic for examining CD13 is not presented.

R21: The analysis of CD13 was initially performed as internal control of expression of ACE2, being CD13 and ACE2 co-expressed in several cell types. However, we agree with the Reviewer that CD13 analysis is unnecessary at this time, therefore we removed CD13 from the manuscript, focusing more on ACE2 and CFTR.

Minor Comments:

1. It should be noted that 16HBE14o- cells are functionally heterozygous due to the insertion of the SV40 sequence within one of the two CFTR sequences. This is misstated in the manuscript.

22: We added this point in the revised manuscript (lines 131-133).

2. Reasons for the apparent regulatory effects of CFTR on other ion channels include its impact on total cell potential difference, a unifying mechanism for many of the ion channels reported.

R23: We added also this point in the revised manuscript (lines 92-94).

3. Given the deleterious effects of inhibiting CFTR on lung health, the concept of inhibiting CFTR as a therapeutic strategy seems quite premature to introduce here. This is particularly true given the substantial effects of SARS-CoV-2 infection on respiratory epithelial, including deleterious effects on mucus clearance.

R24: We agree with the Reviewer. We modified the conclusions being more cautious.

REVIEWER COMMENTS

Reviewer #1 (Remarks to the Author):

The authors thoroughly revised the manuscript and I have no further comments except that the term proliferation (e.g in the abstract) should be replaced by replication.

Reviewer #2 (Remarks to the Author):

The manuscript is largely improved and now makes a much more convincing case, particularly regarding shared biogenesis of ACE2 and CFTR. Nevertheless, there are a few residual critiques in the revised manuscript,

Minor comments

The scatter plots used to derive the Pearson and M1 correlation coefficients should be shown in the supplement (Fig 3, S Fig 2).

Figure 4 does not show evidence of retention of ACE2 in the ER. Biochemical studies or ER protein co-localization would be needed, even if that cellular phenotype were evident, which I do not see in the Figures provided.

Scatter plots are needed in multiple figures, especially those that use individual cell donor as replicate studies. This is lost in some of the alter figures.

Rephrase “accumulation” to more accurately describe mRNA levels: Treatment of Calu-3 cells with pre-miR-145-5p significantly reduced the accumulation of CFTR mRNA (Fig. 6d).

In the results section, Figure 6h-k results are devoid of conclusions on the studies, and do not point out that the authors are observing reduced infection with CFTR inhibition, without effects on CFTR processing. Although reasons for this to occur are spelled out in the discussion, this finding is more speculative compared to the ACE2 – CFTR expression studies in the bulk of the paper.

This sentence in isolation needs editing so it is not taken out of context: “Therefore, it seems that ACE2 expression is mainly regulated by CFTR.”

Verona, 28 September 2022

Ref. NCOMMS-21-44580B – Bezzetti et al.

Point-by-point response letter to the Reviewers

We thank the Reviewers for the positive feedback on our revised version of the manuscript. Our point-by-point responses after the second round of review are marked in green within the text.

Reviewer #1 (Remarks to the Author):

The authors thoroughly revised the manuscript and I have no further comments except that the term proliferation (e.g in the abstract) should be replaced by replication.

R 1.1. The term proliferation has been replaced by replication throughout the manuscript.

Reviewer #2 (Remarks to the Author):

The manuscript is largely improved and now makes a much more convincing case, particularly regarding shared biogenesis of ACE2 and CFTR. Nevertheless, there are a few residual critiques in the revised manuscript.

R 2.1. We agree with the Reviewer that the current manuscript makes a much more convincing case and provides compelling data on the shared biogenesis of ACE2 and CFTR. We thank the Reviewer for her/his helpful suggestions and comments that led to the quality improvement of the current study.

Minor comments

The scatter plots used to derive the Pearson and M1 correlation coefficients should be shown in the supplement (Fig 3, S Fig 2).

R 2.2. Scatter plots showing Pearson and M1 correlation coefficients have been moved to Supplementary information (New Supplementary Fig. 2)

Figure 4 does not show evidence of retention of ACE2 in the ER. Biochemical studies or ER protein co-localization would be needed, even if that cellular phenotype were evident, which I do not see in the Figures provided.

R 2.3. We agree with the Reviewer that ACE2 retention in ER in *CFTR*-null cells (Fig. 4) was not experimentally tested, thus just speculative. We removed that sentence from results section (lines 181-182), highlighting that *CFTR*-null cells only display a remarkable reduction of ACE2 levels, in particular on the apical plasma membrane, as clearly observed in Fig. 4. However, we produced a new figure for addressing the Reviewer's comment. In the Figure Rev. 1 here below, we show that defective *CFTR* channel influences the subcellular localization of ACE2, which is retained into the Endoplasmic Reticulum (ER). This figure is designed for Reviewers' evaluation only.

Scatter plots are needed in multiple figures, especially those that use individual cell donor as replicate studies. This is lost in some of the alter figures.

R 2.4. Most histograms have been replaced by scatter plots, accordingly, but maintaining the bars, in order to facilitate data interpretation by the readers.

Rephrase "accumulation" to more accurately describe mRNA levels: Treatment of Calu-3 cells with pre-miR-145-5p significantly reduced the accumulation of *CFTR* mRNA (Fig. 6d).

R 2.5. The term mRNA accumulation has been replaced by mRNA levels, as suggested by the Reviewer (line 225).

In the results section, Figure 6h-k results are devoid of conclusions on the studies, and do not point out that the authors are observing reduced infection with *CFTR* inhibition, without effects on *CFTR* processing. Although reasons for this to occur are spelled out in the discussion, this finding is more speculative compared to the ACE2 – *CFTR* expression studies in the bulk of the paper.

R 2.6. We pointed out that Figure 6h-k shows a reduced SARS-CoV-2 infection upon *CFTR* functional inhibition, without effects on *CFTR* processing, accordingly (line 234).

This sentence in isolation needs editing so it is not taken out of context: "Therefore, it seems that ACE2 expression is mainly regulated by *CFTR*."

R 2.7. We agree with the Reviewer that that sentence was taken out if context and redundant. Therefore, we removed that sentence.

Figure Rev. 1

Figure Rev. 1 Defective CFTR channel influences the subcellular localization of ACE2, which is retained into the Endoplasmic Reticulum (ER). a) Representative images of immunofluorescence detection of ER resident PDI protein (green) and ACE2 receptor (red) under basal conditions in CFBE41o- (null), CFBE41o- expressing wild-type CFTR (WT) and F508del-CFTR (F508del) cells. Images were acquired with a Zeiss LSM510 confocal microscope (scale bar: 10µm). b, c Colocalization of PDI with ACE2 receptor in ER, tested in the same conditions depicted in panel a, as measured by Manders M1 (b) and Pearson's (c) coefficients.

REVIEWER COMMENTS

Reviewer #2 (Remarks to the Author):

All residual critiques were appropriately addressed. I have no further concerns.

The red/green revision document was helpful in the two stage review - I might adopt this myself.